# Wafer-scale solution-processed 2D material analog resistive memory array for memory-based computing

Baoshan Tang [1], Hasita Veluri[1], Yida Li[1], Zhi Gen Yu [2], Moaz Waqar [3], Jin Feng Leong[1], Maheswari Sivan [1], Evgeny Zamburg [1], Yong-Wei Zhang [2], John Wang [3] & Aaron V-Y. Thean [1✉]

Realization of high-density and reliable resistive random access memories based on two-dimensional semiconductors is crucial toward their development in next-generation information storage and neuromorphic computing. Here, wafer-scale integration of solution-processed two-dimensional $MoS_2$ memristor arrays are reported. The $MoS_2$ memristors achieve excellent endurance, long memory retention, low device variations, and high analog on/off ratio with linear conductance update characteristics. The two-dimensional nanosheets appear to enable a unique way to modulate switching characteristics through the inter-flake sulfur vacancies diffusion, which can be controlled by the flake size distribution. Furthermore, the MNIST handwritten digits recognition shows that the $MoS_2$ memristors can operate with a high accuracy of >98.02%, which demonstrates its feasibility for future analog memory applications. Finally, a monolithic three-dimensional memory cube has been demonstrated by stacking the two-dimensional $MoS_2$ layers, paving the way for the implementation of two memristor into high-density neuromorphic computing system.

---

[1] Department of Electrical and Computer Engineering, National University of Singapore, Singapore 117576, Singapore. [2] Institute of High Performance Computing, Singapore 138632, Singapore. [3] Department of Materials Science and Engineering, National University of Singapore, Singapore 117574, Singapore. ✉email: Aaron.Thean@nus.edu.sg

Analog non-volatile memory devices capable of multi-states like memristors, promise to enable new classes of energy-efficient computation like in-memory computing[1,2] and neuromorphic computing[3,4] to disrupt conventional graphics processing unit (GPU)-based neural-network accelerators. In this case, memory devices assume compute roles beyond just data storage. As such, memory device endurance and variability comparable to logic devices are now desired. However, endurance and variability for analog resistive random access memory (RRAM), especially for resistive switching devices based on traditional amorphous and polycrystalline metal oxides continues to suffer material-level challenges due to the unavoidable tradeoffs between defect stability and defect recovery, forcing limited optimization window between endurance, programming voltage, and memory retention[5].

Two-dimensional (2D) materials with a wide variety of electronic properties including an expanded range of vacancy activation energies offer a new palette to engineer the switching materials and their defects for resistive memories[6]. With improved large-scale growth techniques and the compatibility with CMOS integration, emerging 2D materials have attracted increasing attention in memristive devices[7–10]. Recent studies have revealed that 2D materials, with their unique edge and layered properties, can be more accurately tuned to enhance their switching characteristic not seen in oxide-based memristive devices[11,12]. Several intriguing performance milestones have been achieved so far in 2D memristive devices, eg., sub-pA current with femtojoule per bit energy consumption[13]; operation up to 50 GHz in radio frequency switches[14]; switching thresholds approaching 100 mV[15]; stable operation up to 340 °C[11] and switching in sub-nanometer thickness[16,17]. However, most of these reports are based on isolated device without indication of its viability in large-scale arrays. Despite $10 \times 10$ memristive crossbar arrays being demonstrated by chemical vapor deposition (CVD) $h$-BN[18,19], the high growth temperature and post-synthesis transfer process increase the integration complexity, thus impeding their implementation in large-scale circuit application. With low process temperature and compatibility with high-precision optical lithography patterning, solution-processed 2D materials offer a practical approach to co-integrate 2D material with Si CMOS to enhancing future on-chip computational functionality.

The liquid-exfoliated 2D nanosheets retain pristine crystal quality and clean van der Waals interfaces, thus ensuring excellent charge transport[20,21]. Analogous to the grain boundaries defects in CVD-grown 2D materials[22], the edge defects in the liquid-exfoliated nanosheets assist the interlayer diffusion of conductive filament (CF) and present an efficient pathway for resistive switching (RS) modulation. This is further enhanced by the ability to control the size distribution and edge defects density of the liquid exfoliated 2D nanosheets[23–27]. Despite the promises, the solution-processed 2D memristors devices reported to date are still challenged by poor endurance, low yield, and large device-to-device variation[25,28,29].

In this work, we demonstrate a reliable and scalable approach to fabricate memristor arrays at wafer-scale. Through solution processing and spin-coating on wafer, we produce a continuous thin-film network of monodispersed $MoS_2$ nanosheets. Remarkably, the $MoS_2$ memristor exhibits forming-free switching with a high endurance of $1 \times 10^7$ cycles, low device-to-device variability (19.7% for set and 18.5% for reset), excellent retention for 10 years and a remarkable wafer-scale crossbar arrays scalability. In-depth materials and property characterization reveal that the RS characteristics of the 2D $MoS_2$ memristors are modulated by the sulfur vacancies ($V_S$) percolation along the flake edges. As a demonstration, we implemented a 3-layer convolutional neural network (CNN) model using the $MoS_2$ memristors for the recognition of MNIST handwritten digits. With excellent switching linearity and low variation, our CNN model achieved high recognition accuracy of 98.02%. Furthermore, we proposed a 3D memory cube through layer-by-layer stacking the 2D $MoS_2$ nanosheets in a transfer-free manner, opening a promising pathway for building 3D integrated circuits with 2D materials.

## Results

**Production of $MoS_2$ nanosheets dispersion**. The $MoS_2$ bulk crystal was exfoliated into nanosheets by electrochemical intercalation followed by a mild sonication as illustrated in Fig. 1a (see **Method**)[20,21]. This process creates a dark green $MoS_2$ suspension (Fig. 1b inset). The optical UV-Vis absorption spectroscopy of the obtained $MoS_2$ suspension exhibits two excitonic peaks near 678 nm and 614 nm, suggesting the high-quality semiconducting $MoS_2$ nanosheets[30,31]. Raman spectroscopy indicates the existence of monolayer to few-layer $MoS_2$ nanosheets (Fig. 1c)[32,33]. Distinct peak positions and high intensity at around 678 nm and 614 nm in photoluminescence spectroscopy (PL) are also observed in exfoliated $MoS_2$ monolayers, consistent with the UV-Vis absorption spectroscopy, further confirming that the intrinsic electronic properties of $MoS_2$ are preserved (Supplementary Fig. S1). Based on atomic force microscopy statistics, the exfoliated $MoS_2$ nanosheets are about 2.6 nm thick on average, corresponding to few-layer $MoS_2$ (Fig. 1d). By cascade centrifuging[20], three batches of $MoS_2$ suspensions are obtained with narrow lateral size distribution centered at 0.48 μm, 1.20 μm, and 2.40 μm, denoted as suspension A, B and C, respectively (Fig. 1e). X-ray photoelectron spectroscopy (XPS) of the exfoliated $MoS_2$ nanosheets corroborated that the $MoS_2$ nanosheets exhibit pristine chemical states with no oxidation (Supplementary Fig. S2). The stoichiometry of $MoS_2$ with Mo/S ratio of 1:1.93 is determined from the XPS results, corresponding to $MoS_2$ with sulfur deficiency. The above experimental observations validate the successful exfoliation of the bulk $MoS_2$ crystals into few-layered sub-stoichiometric $MoS_2$ nanosheets laced with $V_S$.

**Fabrication and characterization of the $MoS_2$ Film**. Using the $MoS_2$ nanosheets suspension, we prepared uniform $MoS_2$ thin film on standard 2-inch $Si/SO_2$ wafer via a simple spin coating process, achieving a smooth and continuous surface with roughness of 1.2 nm and average thickness ranging from 10.5 to 11.4 nm (Fig. 2a, Supplementary Fig. S3, 4). The smooth and continuous surface plays a significant role in the enhancement of device yield and reduction of the performance variability. Raman mapping of the patterned letters 'NUS' (Fig. 2b–d) together with identical Raman spectra at random spots on the $MoS_2$ film (Fig. 2e) further confirm the superior uniformity and full coverage of the $MoS_2$ film. With 2D layered morphology, the ultrathin $MoS_2$ nanosheets can evenly and tightly overlap with each other to form a continuous thin film with reduced areal density of grain boundaries within the horizontal plane as evidenced by the scanning electron microscopy image (Fig. 2f).

The obtained $MoS_2$ thin film possesses high crystallinity with perfect hexagonal atomic arrangement of Mo and S atoms in $MoS_2$ basal planes (Fig. 2g), typical of a high-quality $MoS_2$ flake. In addition, the tightly stacked $MoS_2$ layers in the vertical direction have been confirmed by the cross-sectional scanning transmission electron microscopy (STEM) imaging and corresponding electron energy loss spectroscopy (EELS) mapping. As shown in Fig. 2h, the $MoS_2$ stacking layers exhibit large-area plane-to-plane contact with nearly atomically flat 2D interfaces, ensuring efficient charge transport across the $MoS_2$ stacks. EELS mapping of the $MoS_2$ layers demonstrates the preservation of its

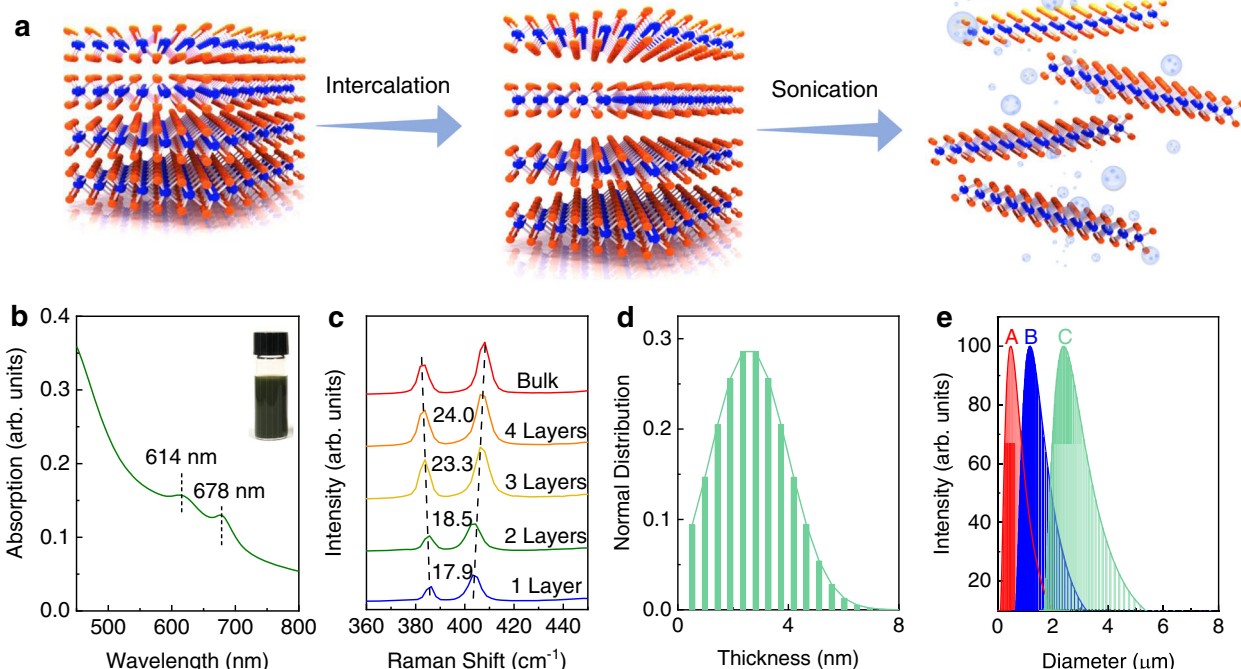

**Fig. 1 The liquid exfoliation of MoS₂ Nanosheets. a** Schematic illustration of the liquid exfoliation process of bulk MoS₂ crystal. **b** UV-vis absorption spectrum of MoS₂-IPA solution. Inset is the photograph of the MoS₂-IPA dispersion. **c** Raman spectra of exfoliated MoS₂ nanosheets. Raman signal of bulk MoS₂ crystal is also presented for comparison. **d** Atomic force microscopy statistics indicating the flake thickness distribution of the MoS₂ dispersions. **e** The lateral flake size distribution of three types of MoS₂ dispersions. To be noted, arb. units refer to arbitrary units.

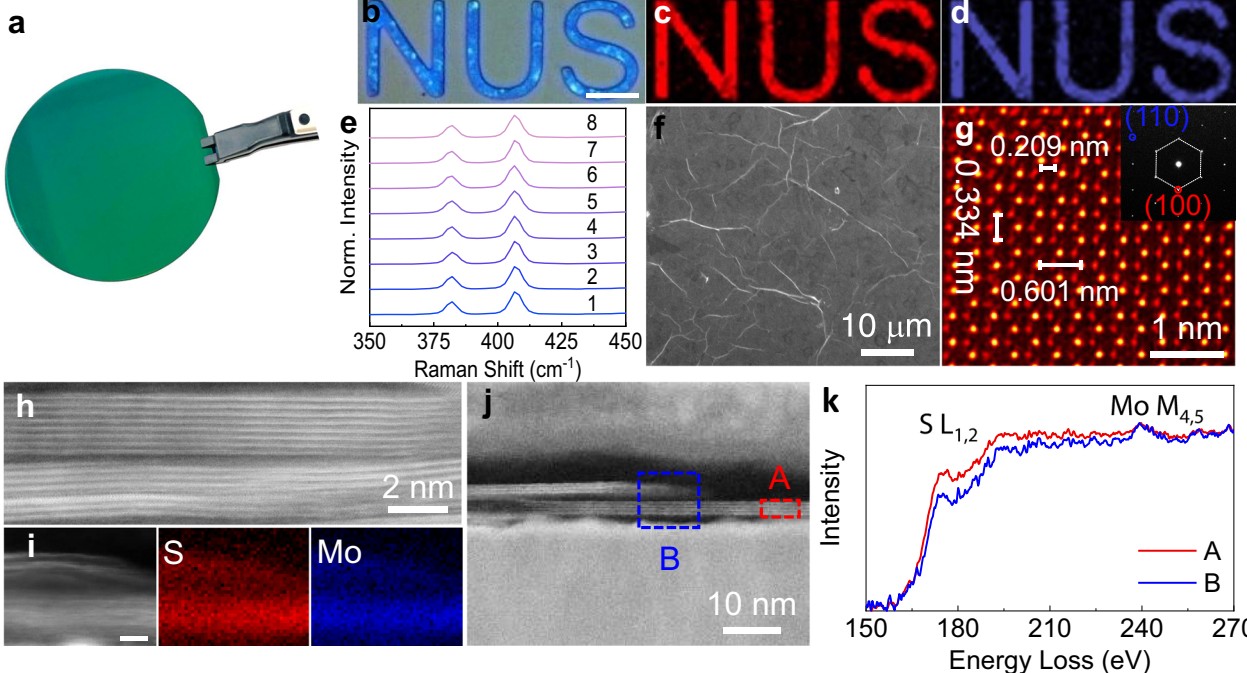

**Fig. 2 Characterization of spin-cast MoS₂ thin film. a** Optical image of the wafer-scale MoS₂ thin film. **b** optical image of patterned MoS₂ thin film and corresponding Raman mapping of $E^1_{2g}$ mode (**c**) and $A_{1g}$ mode (**d**). The scale bar in (**b**) is 10 μm. **e** Raman spectra collected on random spots from **a**. **f** SEM image of the MoS₂ thin film. The wrinkles are the edges of MoS₂ nanosheets. **g** Atomic resolution high-angle annular dark field (HAADF) STEM image of MoS₂ nanosheets (false-colored). **h** Cross-sectional HAADF-STEM image and **i**. corresponding elemental mapping of MoS₂ thin film. **j** The cross-sectional ADF-STEM image taken at the junction region of the MoS₂ thin film. **k** EELS spectra of region A and B taken from **j**.

intrinsic chemical composition without noticeable oxidation due to the room-temperature processing involved in materials preparation and device fabrication (Fig. 2i and Supplementary Fig. S5). Figure 2j shows the junction region of the two $MoS_2$ stacking layers. Two EELS spectra of the S-$L_{1,2}$ and the Mo-$M_{4,5}$ edges have been obtained exactly at the junction site (region B) and away from the junction site (region A), respectively (Fig. 2k). It clearly shows that region B has a weaker intensity in S-$L_{1,2}$ edge, indicative of the sulfur deficiency at the junction region of $MoS_2$ stacks. Furthermore, as an electron donor, $V_S$ can effectively tune the work function of $MoS_2$. As shown in Supplementary Fig. S6, the peripheral regions of $MoS_2$ nanosheets possess distinctly more negative potential than the central region, corresponding to lower work function induced by $V_S$. The defective edges serve as a vertical percolation path for $V_S$ across the $MoS_2$ stacks, offering a unique approach for the RS modulation of the solution-processable $MoS_2$ memristor by engineering their flake sizes.

**Electrical characterization of the solution processable $MoS_2$ memristor.** In order to gain insight into the underlying transport mechanism associated with $V_S$ in the $MoS_2$ switching layer, RS switching characteristics of a single $MoS_2$ flake are examined. As shown in the conductive AFM (C-AFM) measurement (Supplementary Fig. S7), a large hysteresis window at low voltage in the $I$-$V$ curve is observed at the flake edges, while being absent at the center of the flakes, thus revealing the importance of the $V_S$ in the RS phenomenon. To be noted, the observed hysteresis in the C-AFM experiment shows the stable and repeatable RS behavior of the $MoS_2$ memristors even in scaled devices down to 10 nm (tip size), demonstrating their ability for high-density memory integration. Furthermore, by applying constant voltage stress on the $MoS_2$ memristors, we observe the characteristic random telegraph signal with two discrete conductance states, revealing the charge trapping and de-trapping related to $V_S$ (Supplementary Fig. S8). The sweep rate-dependent set/reset threshold voltages imply that the memory effect of $MoS_2$ memristors is dominated by nanoionics transport mechanism (Supplementary Fig. S9)[16], further confirming the significant role that $V_S$ plays in the RS conduction mechanism in $MoS_2$ memristor.

Considering that the $V_S$ are confined at the edges of the $MoS_2$ nanosheets, a strong correlation between $MoS_2$ nanosheet size and the corresponding RS characteristics is expected. To validate that assumption, memristors with different $MoS_2$ nanosheet sizes are fabricated (Supplementary Fig. S10). Their typical $I$-$V$ characteristics are shown in Fig. 3a–c. All $MoS_2$ memristors exhibit forming-free switching characteristics, promoted by the presence of $V_S$ in the exfoliated $MoS_2$ nanosheets, which would be favorable to produce smooth dielectric breakdown[18,22,34–36]. A clear bipolar switching with stable low resistance state (LRS) and high resistance state (HRS) is observed. Due to the presence of edge-confined $V_S$, an anomalous nanosheet-size dependent RS characteristic have been observed, where the shrinking in the average nanosheets size ($\lambda$) results in a reduced set/reset voltage ($V_{set}$/$V_{reset}$) and their cycle-to-cycle variations. Specifically, the memristor with the smallest $MoS_2$ nanosheets exhibits the lowest $V_{set}$ (0.65 V)/$V_{reset}$ (−0.90 V) and cycle-to-cycle switching voltage variations. The discrepancy in the switch voltage arises from the difference in nanosheet size-related $V_S$ defect density. Consequently, smaller nanosheets are expected to have higher $V_S$ density attributed to increased edge-to-basal plane ratio, implying the possibility of achieving ultra-low switching voltage by modulating the nanosheet size. Meanwhile, the nanosheet size plays an important role in the reduction of the cycle-to-cycle switching voltage variations. This is attributed to more uniform

average $V_S$ percolation length under the circumstance of smaller $MoS_2$ nanosheets (Supplementary Fig. S11). Small $MoS_2$ nanosheets are expected to provide a smoother and more uniform pathway for $V_S$ percolation while larger flakes would produce more tortuous channels. The statistical analyses of 50 devices under different flake sizes further confirm the flake size dependent RS characteristics in $MoS_2$ memristor (Supplementary Fig. S12). In order to better understand the geometric effect of the $MoS_2$ nanosheets on the RS characteristics, the diffusion energy landscape of $V_S$ is further explored by the density functional theory (DFT) calculations (details in Supplementary Note I, Figure S13). Figure 3d shows the impact of the $MoS_2$ molecular sizes on the $V_S$ migration barriers. Clearly, decreasing the $MoS_2$ molecular size results in the reduction of the diffusion energy barrier for $V_S$, reaching a small $V_S$ diffusion energy barrier of 0.75 eV, which is consistent with the reported behavior of polycrystalline $MoS_2$ memtransistor[37]. Supported by the DFT calculations, the kinetics of the $V_S$ diffusion reveals that the nanosheet size effect is highly related to the $V_S$ diffusion barrier along the nanosheets edges, representing an effective way for the engineering of RS in $MoS_2$ memristor.

Given that the migration of $V_S$ is a thermodynamic process, it is expected that the $MoS_2$ memristor should be influenced by temperature. Temperature dependent $I$-$V$ sweeps at HRS and LRS states have been performed as shown in Fig. 3e, f. For both LRS and HRS profile, the current shows a non-linear relationship with voltage and increases with the increase of temperature, suggesting the existence of Schottky barrier. By considering different transport models, the $I$-$V$ relationship under HRS and LRS states are well-fitted with Schottky emission model as a linear dependence of $J$ on $E^{1/2}$ is obtained, where $J$ is the current density and $E$ is the electric field (Fig. 3g, h). The Schottky emission function is expressed as below:

$$J \propto A^* T^2 \exp[-q(\varnothing_B - \sqrt{qE/4\pi\varepsilon_r\varepsilon_0}/(kT)]$$

where $T$ is the absolute temperature, $q$ is the electronic charge, $A^*$ is the effective Richardson constant, $\varnothing_B$ is the Schottky barrier height. $k$ is the Boltzmann's constant, $\varepsilon_0$ is the vacuum permittivity, and $\varepsilon_r$ is the optical dielectric constant. From the Schottky emission equation, the barrier height for HRS is calculated as high as 500 meV, whereas, the barrier height is greatly reduced to only 93 meV in LRS state. The double-logarithmic plots of the $I$-$V$ curve of $MoS_2$ memristor shows that the HRS state follows trap-associated space-charge limited conduction (SCLC) theory, while the LRS state is governed by Ohmic conduction behavior which is caused by the formation of conductive filaments (Supplementary Fig. S14). At HRS states, since the height of the Schottky barrier is very large, only few electrons can be injected from the metal to the semiconductor in $MoS_2$ because the thermionic current exponentially decreases with the increasing of barrier height. In LRS state, the current gradually increases with increasing temperature, showing semiconductor-like behaviors, excluding the metal conductive filaments (Fig. 3e, g). By plotting ln $I$ a function of $T^{-1/4}$ (Fig. S14), the transport characteristics can be well fitted with the Mott-Variable range hopping model (Mott-VRH) at temperature above 110 K[38,39]. This suggests that the electrons hop through the conductive filament composed of $V_S$ in the LRS states.

Based on the above observation, the conduction mechanism of the $MoS_2$ memristor is schematically illustrated in Fig. 3i, j. At HRS, the conduction mechanism follows the SCLC conduction. When positive voltage is applied to the top contact (Ti), the positively charged $V_S$ diffuse along the $MoS_2$ nanosheet edges towards bottom contact (Pt). As $V_S$ approaching the Pt electrode, the Schottky barrier height is reduced at the Pt-$MoS_2$ interface.

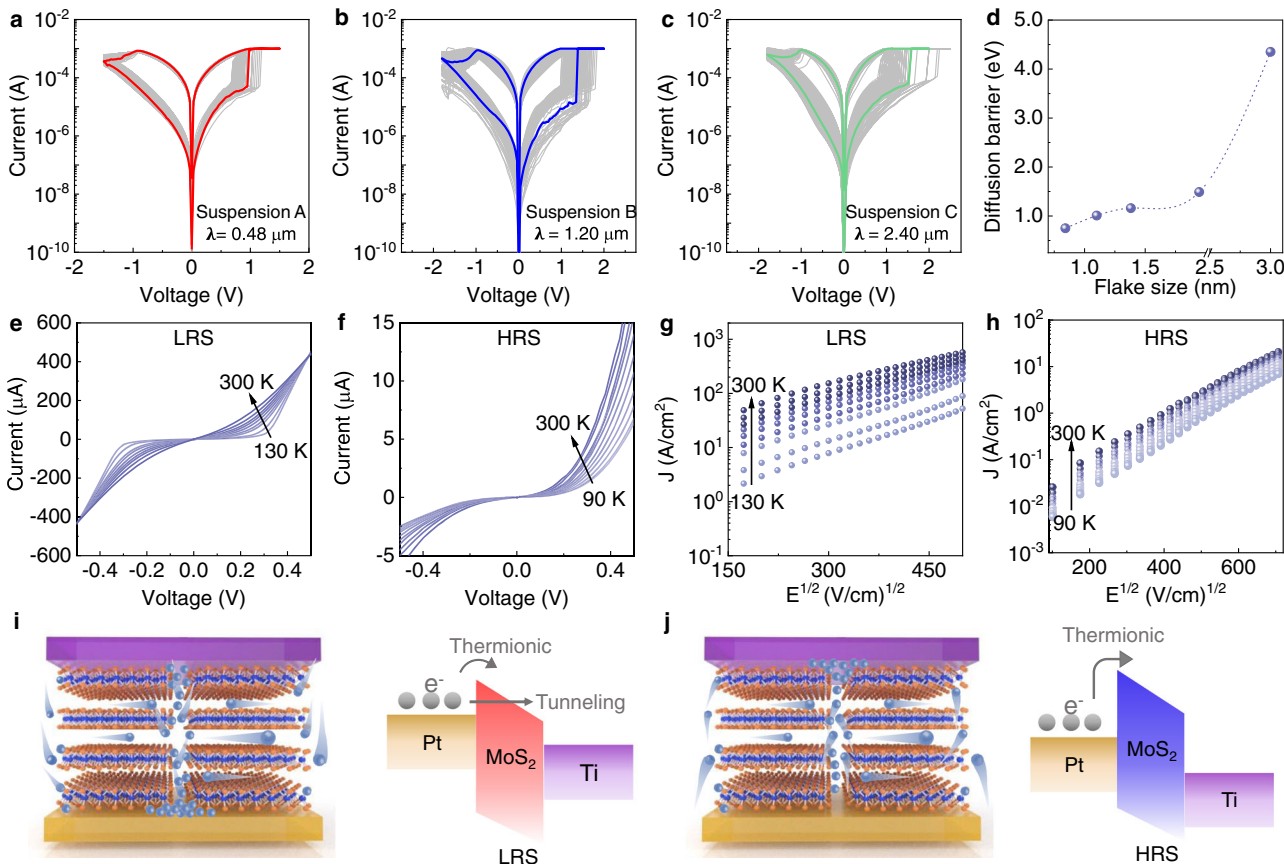

**Fig. 3 Electrical performance characterization of the MoS₂ memristors. a–c** 200 representative *I-V* curves of MoS₂ memristors made from suspensions A, B, and C, respectively. To be noted, the electrode size is 5 × 5 μm. **d** Calculated $V_S$ diffusion barrier energy versus flake size. *I-V* characteristics of MoS₂ memristor measured at different temperatures in LRS (**e**) and HRS (**f**). Schottky emission fitting for LRS (**g**) and HRS state (**h**). Schematic diagrams of the set (**i**)/reset (**j**) process and corresponding interface band alignment. The light blue balls represent $V_S$.

The continuous positive bias causes a large accumulation of $V_S$ and eventually results in the formation of $V_S$ conductive filaments, bridging the Ti and Pt electrode. The device is set to the LRS state (Fig. 3i). At LRS state, the conduction mechanism follows the Mott-VRH model, where electrons hop through the conductive filament composed of $V_S$. When a negative bias is applied to the top electrode, $V_S$ are extracted backwards and the conductive filaments rupture (Fig. 3j). The device is reset to the HRS state. Meanwhile, the depletion of $V_S$ at the Pt-MoS₂ interfaces leads to the increase in the Schottky barrier associated with reduced current conductance. Due to the migration of $V_S$ under voltage bias, we observe a dynamic modulation of the Schottky barrier near the MoS₂-Pt interface with 500 meV and 93 meV under HRS and LRS, respectively. Overall, the conduction mechanism in the MoS₂ memristors is dominated by the formation and rupture of the conductive filaments due to the percolation of $V_S$ along the nanosheet edges. Unlike the stochastic formation of CF in amorphous oxide-based memristor, the solution-processed MoS₂ memristor enables a better control over RS characteristic due to the unique edge-confined $V_S$ conduction mechanism. Therefore, it provides an effective engineering way for the modulation of the RS characteristics via controlling the size of the MoS₂ nanosheets, not possible in conventional oxides-based memristors[40].

**Implementation of CNN with the solution-processed MoS₂ memristor.** The MoS₂ memristor shows robust and reliable switching characteristics as supported by their excellent endurance and retention. As shown in Supplementary Fig. S15, the time-dependent resistance exhibits little change in ON and OFF state for $1 \times 10^5$ s without degradation and demonstrates extrapolated 10-year retention at 85 °C. Moreover, the repeatability has been studied through setting/resetting the device by voltage pulses up to $1 \times 10^7$ cycles (Fig. 4a). The resistances at the LRS and HRS extracted from different pulse cycles show record-high endurance, even comparable to the typical endurance of commercial flash memories[41]. The uniform bipolar RS behavior can also be preserved when the device size is reduced to 100 nm × 100 nm (Supplementary Fig. S16), demonstrating good scalability of the proposed process. Given the superior uniformity of the MoS₂ thin film and reliable performance in our MoS₂ memristors, the wafer-scale MoS₂ memristors arrays (Fig. 4b inset and Supplementary Fig. S17) were fabricated. To address the device-to-device reproducibility, the distributions of $V_{set}$ and $V_{reset}$ obtained by measuring 73 devices over the entire wafer has been done. As shown in Fig. 4b, the variations (defined as $\sigma/V_{mean}$) for set and reset voltages are calculated to be only 19.7% and 18.5%, which are much smaller as compared to other solution-processed 2D memory devices[28]. Furthermore, the conductance of a 6 × 6 MoS₂ memristor array was programmed into displaying the number "2" and alphabet "D"; spatially representing the analog conductance states across the array, over a period of 1 h without losing its programmed state (Supplementary Fig. S18).

The dynamic response of the solution-processed MoS₂ memristors have been studied under pulsed electric stimuli with different amplitudes, durations and intervals. Overall, an accurate CNN training requires multibit analog resistive states and symmetric conductance change. Therefore, we characterized the

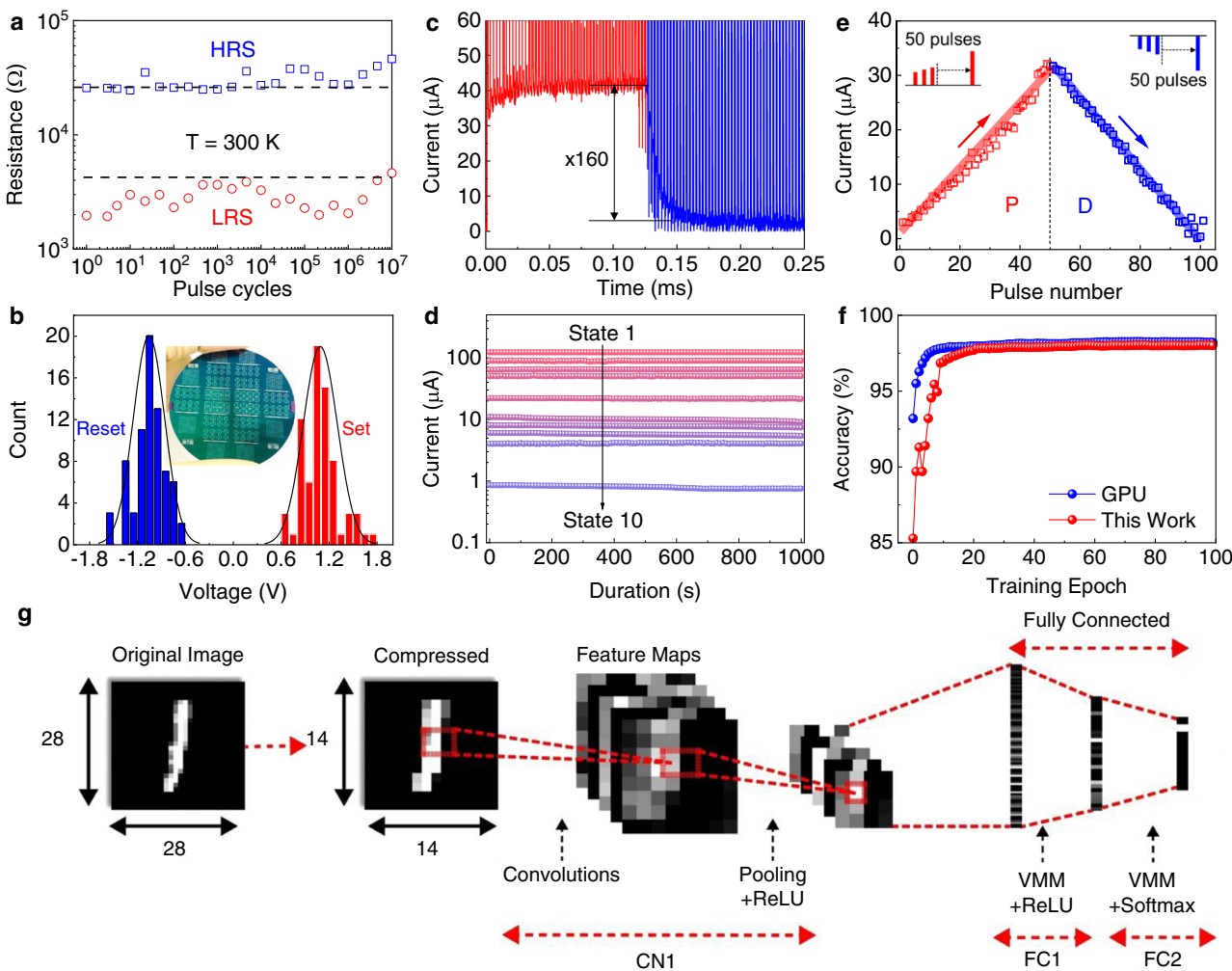

**Fig. 4 Dynamic response of the MoS$_2$ memristor for memory-based computing. a** AC endurance obtained with 1 μs pulse width (1.5 V for set, and -2.0 V for reset). **b** Statistical distribution of the set/reset voltages for 73 devices measured over the entire wafer. The inset is the MoS$_2$ memristors on a 2-inch wafer. **c** Potentiation and depression of MoS$_2$ memristor by sequence of pulse train showing high analog on/off conductance ratio. **d** I-t measured at 0.2 V after sequentially resetting the MoS$_2$ memristors into 10 memory states. **e** Conductance update as a function of incremental potentiation and depression pulse numbers. **f** Comparison of classification accuracy of a 3-layer CNN executed in-memory with GPU execution over 100 epochs. GPU execution achieved 98.24% accuracy while proposed technique achieved a similar accuracy 98.02%. **g** A 3-layer DCNN with 1 convolutional and 2 fully-connected layers used to classify handwritten digits. The convolution layer consists of 5 7 × 7 filters while the hidden node in the fully connected layers has 30 neurons. Notably, MoS$_2$ memristor with flake size of 0.48 um was examined in this section.

analog conductance response by applying sequence of pulse train consisting of set (1.5 V, 1 us), reset (−2 V, 10 us) for potentiation-depression (P-D) pulses and read pulses (0.1 V, 1 us) after each P-D pulse. As shown in Fig. 4c, the conductivity of the MoS$_2$ memristors show analog potentiation and depression between different resistive states with a remarkably high on/off ratio of 160. The high analog on/off ratio in our MoS$_2$ memristor is beneficial for accessing multiple synaptic weight values in neural network algorithms[40,42]. Each of the ten selected conductive states shows no degradation in conductance over time, indicating excellent multilevel retention capability of the MoS$_2$ memristor (Fig. 4d). The multiple memory states are programmable within single device, (Supplementary Fig. S19), and further demonstrated to be highly reproducible among different devices (Supplementary Fig. S20). Despite our MoS$_2$ memristor showing high analog memory states, the weight updates in an asymmetry way, especially at the start of the P-D cycle where the MoS$_2$ memristor conductance abruptly arises and decays. The abrupt weight update can be ascribed to the fast switch time in our MoS$_2$ memristors due to mobile V$_S$ with reduced diffusion barrier

energy. Indeed, as shown in Supplementary Fig. S21, the MoS$_2$ has a fast switch time of 40 ns at a pulse amplitude of 2.0 V. Therefore, to avoid the abrupt potentiation and depression and reach a linear weight update, an incremental voltage operating scheme with a shorter pulse width and lower amplitude (potentiation: 1.0 V to 1.75 V with 15 mV steps, pulse width 100 ns; depression: −1.5 V to −2.25 V with steps of 15 mV; pulse width 500 ns) was applied to the MoS$_2$ memristor. As a result, symmetric conductance potentiation and depression with nearly linear modulation have been achieved, with an on/off conductance ratio of 10, which is essential to implement reliable low-complexity and low-energy analog in-memory matrix multiplication (Fig. 4e)[43–45]. To estimate the effect of the MoS$_2$ memristor on the pattern recognition accuracy, we performed a three-layer CNN simulation based on the MNIST handwritten data set (See details in Supplementary Note II, Figs. S22–S24). Considering the size of the RRAM-array data, the 28 × 28-pixel raw images are compressed into smaller images with 14 × 14 pixels and processed as shown in Fig. 4g. After training the 3-layer deep convolutional neural network (DCNN) implemented

on MoS$_2$ RRAM array with 50,000 images using the in-memory computation technique, we achieved a high accuracy of 98.02% (Fig. 4f and Supplementary Table I). Supplementary Table II benchmarks the overall performance metrics of different 2D materials-based memristors as well as the conventional oxide-based memristors. Our MoS$_2$ memristor shows the best performance in the integration size, endurance, learning accuracy and number of conductance states relative to other 2D materials. Especially, our proposed processes are transfer-free and room-temperature based, offering great compatibility with thermal budget limited 3D monolithic integration as well as flexible electronics. Moreover, our devices feature an alternate process dimension of performance modulation via flake size engineering, which is lacking in current oxide-based RRAMs. Despite limitations in high switching power and relatively large device variations of the current design, we believe there is still significant opportunity for improvement by materials, devices and circuits co-optimization in the future.

**Demonstration of monolithic 3D memory cube.** Monolithic 3D integration (M3D) of memory and logic component leads to high-density device, providing a promising avenue to address the conventional memory wall bottleneck[46,47]. However, for M3D integration of 2D materials, the challenges lie in the high-temperature growth and post-synthesis transfer. On the one hand, the CVD growth process requires high temperature that violates the M3D thermal budget. In M3D, the process temperature of the upper tiers should not exceed a critical temperature of 500 °C, above which, back-end-of-line degradation, silicide deterioration and dopant diffusion in the lower tiers will take place[48]. On the other hand, the post-growth transfer of 2D film with the assistance of polymer film often suffers from problems like film cracks, surface wrinkles and contamination from polymer residues. Those problems often lead to large performance variation and even failure of the devices. More importantly, the transfer process increases the integration complexity, especially for M3D integration, thus impeding their application in large-scale manufacturing. In contrast, solution processing of 2D materials bears several advantages including room temperature processing, substrate agnosticism, low fabrication cost and wafer-scale scalability, thus representing an ideal platform for M3D integration with 2D materials.

As a proof of concept, a three-layered memory stack has been fabricated via sequential deposition of Ti, Pt and spin-coating MoS$_2$. The overview of the 3D stacked MoS$_2$ memristor and the corresponding electrical measurements can be found in Supplementary Fig. S25. To study the structure of the 3D stacked MoS$_2$ memristor, cross-sectional TEM and energy-dispersive X-ray spectroscopy (EDS) were carried out. As shown in the Fig. 5a, the 3D stacked MoS$_2$ memristor features a simple configuration in which the adjacent memory layers have the shared contact, demonstrating the successful fabrication of the 3D stacked MoS$_2$ memristor. The MoS$_2$ layers in the top, middle and bottom memristor show similar thickness of around 10 nm, which indicates the controllability of the deposition process and excellent uniformity of the solution-processed MoS$_2$ film. A high-resolution cross-section TEM taken in the middle layer shows the intimate contact between MoS$_2$ and electrodes, ensuring the efficient charge transport and intactness of the structure (Fig. 5b). The delicate 3D structure has been further elaborated by the elemental mapping, which shows the alternative distribution of Pt, Ti, and MoS$_2$ in sequence (Fig. 5c). Each representative layer of the MoS$_2$ memory stack can be accessible and programmable independently, exhibiting stable and bipolar switching characteristics analog to their planar memristor

counterpart (Fig. 5d, Supplementary Fig. S25d–f). Ultimately, a monolithic 3D circuit has been proposed based on the solution-processable 3D MoS$_2$ memristors. Figure 5e depicts a vertically stacked memory architecture with solution-processed 2D MoS$_2$ thin film as the active switching layer, sandwiched between metal lines of top and bottom electrodes. Etch node at the cross-point of the arrays represents an individual memory cell. Benefits provided by room-temperature process, the storage nodes are vertically stacked in a facile manner through a sequential spin-coating of 2D MoS$_2$ and deposition of metal contacts. Based on our previous reports on M3D integration[26,49], inter-layer dielectric between each layer of metal lines and selectors at each node of cross-point can be readily integrated in our monolithic 3D circuits to minimize the cross-talk issue and accurately program each memory cell. As an example, the M3D integrated one transistor one RRAM (1T1R) arrays with corresponding circuit diagram are illustrated in Supplementary Fig. S26. Our demonstration shows that the solution-processed 2D materials can be 3D monolithically integrated in back-end-of-line (BEOL) for high density embedded memory for storage and analog computing.

## Discussion

In summary, we demonstrate wafer-scale memristor arrays by using solution-processed MoS$_2$ nanosheets. The solution-processed MoS$_2$ memristor arrays exhibit robust and reliable performance with excellent endurance, low device-to-device variation, linear weight updates and high analog on/off ratio. Materials characterization and electrical measurement reveals that the migration of V$_S$ along the edges of the MoS$_2$ nanosheets play a critical role, which provides a flake-based way for the RS modulation. The transfer-free processing of 2D layers at room temperature has the potential to be scaled up for mass production, and enable their integration on flexible substrates, thus providing a versatile platform for flexible and wearable electronics. With the excellent properties, the solution-processed MoS$_2$ memristor arrays can achieve 98.02% pattern recognition accuracy. Furthermore, a fully functional 3-layered memristors have been demonstrated based on the solution-processed 2D memristors, which provides a promising strategy for the M3D integration of 2D materials. The present work opens a door for large-scale and reliable memory integration based on 2D materials for neuromorphic computing implementation.

## Methods

**Preparation of MoS$_2$ nanosheets dispersions.** Typically, MoS$_2$ single crystal was intercalated with tetraethylammonium bromide (Sigma-Aldrich) solution in acetonitrile (5 mg mL$^{-1}$). After the electrochemical intercalation, the MoS$_2$ crystal was rinsed with isopropyl alcohol (IPA) three times, followed by ultrasonication in dimethylformamide (DMF). The MoS$_2$ dispersion was subsequently centrifuged at 1000 rate per minute (r.p.m) for 5 min to remove the unexfoliated particles. To prepare nanosheets with different sizes, the supernatant was collected and subjected to a second sonication for 6 h. Subsequently, the dispersions were centrifuged at centrifuge speeds of 1.0 krpm for 5 min. The sediment was collected and re-dispersed in IPA (suspension C). The supernatant was subjected to further centrifuge at 2.0 krpm. Again, the sediment was collected and re-dispersed in IPA (suspension B). The final supernatant was collected and named suspension A. To prepare suspension D, suspension A was further sonicated, followed by centrifuge at 5000 r.p.m for 3 min. The supernatant was collected and named suspension D.

**Fabrication of RRAM devices.** To make the device, the MoS$_2$ nanosheets dispersion was directly spin-coated onto Si/SiO$_2$ substrate with pre-patterned bottom contact of Ti/Pt (30 nm/30 nm) at room temperature. After that, Ti contacts (30 nm) were deposited capped with Pt (30 nm) by e-beam evaporator to form the Ti/Pt-MoS$_2$-Ti/Pt crossbar devices.

**Characterization.** Room-temperature electrical measurement was conducted in a four-probe station connected to semiconductor parameter analyser (Agilent B1500A) and a waveform function generator (B1530, Agilent). Varied temperature

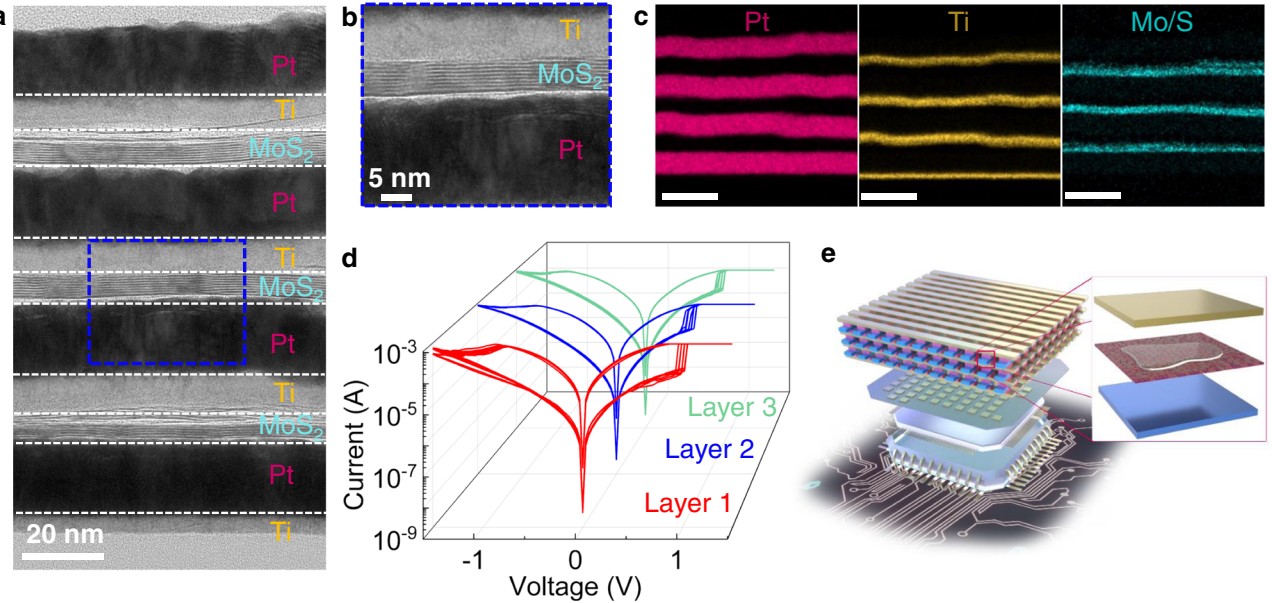

**Fig. 5 Demonstration of 3D stacked 2D MoS₂ memristors.** Cross-sectional TEM images of the full stack (**a**) and middle layer (**b**) of MoS₂ memristor. **c** Elemental mapping of the 3D stacked MoS₂ memristor. **d** *I-V* characteristics of 3D stacked MoS₂ memristors, showing stable and bipolar resistive switching at each layer. **e** Schematic diagram of 3D memristor array with buried metal interconnects and logic circuits.

electrical measurement was conducted in a Lakeshore cryo-probe station. During all electrical measurements, the bottom electrode was grounded and top electrode was biased. Optical images were captured with Olympus BX 51M microscope. AFM images were taken with Bruker Dimension Icon in tapping mode. The surface chemistry of the samples was examined with an XPS in a VG ESCALAB 220i-XL system using a monochromatic Al K$_\alpha$ source at the pass energy of 10 eV for high measurement resolution. Raman spectra were obtained on a Renishaw Raman microscope with a 514 nm excitation and a 100× objective. The laser power was kept below 1 mW to avoid damage. The Si peak at 520.7 cm$^{-1}$ was used for calibration in the data analysis of Raman and PL spectra. Kelvin probe force microscope (KPFM) was performed using a Park atomic force microscope under ambient conditions. A Si cantilever tip coated with Pt-Ir (SCM-PIT, Bruker Co.) was used in the tapping mode. Electrical contacts to the cantilever were grounded during the measurements. An AC voltage of 2 V was applied to the tip while the tip height was kept constant at 5 nm. The STEM and EELS studies were conducted using a JEOL ARM200F atomic resolution analytical electron microscope operating at 200 kV equipped with a cold field-emission gun and a new ASCOR 5th order aberration corrector and a Gatan Quantum ER spectrometer. For qualitative comparison, both the EELS spectra have been normalized to the Mo peak.

**Calculation details**. All calculations were carried out using the density functional theory (DFT) with the generalized Perdew-Burke-Ernzerhof (PBE)[50] and the projector augmented-wave (PAW) pseudopotential plane-wave method[51] as implemented in the Vienna Ab initio Simulation Package (VASP) code[52]. For the PAW pseudopotential of Mo, we included 4d$^4$ and 5s$^2$ as valence; For S, the $n = 3$ shell was included as valence (3s$^2$ and 3p$^4$). A $12 \times 12 \times 1$ Monkhorst-Pack k-point grid was used for monolayer unit cell geometry optimization and a plane-wave basis set with an energy cut-off of 500 eV was adopted. The optimized unitcell was used to build the multilayer MoS₂ flake with different sizes. In this study, we carried out calculations with the van der Waals correction by employing optB88-vdW functional[53]. Good convergence was obtained with these parameters and the total energy was converged to $1.0 \times 10^{-6}$ eV per atom. No spin polarization was considered in this study. The energy minimization was performed using the limited memory BFGS method. The climbing-image Nudged Elastic Band (NEB) method[54] was used to figure out the diffusion of S vacancy in the minimum energy landscape and energy barriers.

## Data availability
The authors declare that the data that support the findings of this study are available within the article and its Supplementary Information files. All other relevant data are available from the corresponding author upon request.

## Code availability
All relevant code or algorithm are available from the corresponding author upon request.

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

## Acknowledgements

This work is supported in part by Singapore's National Research Foundation grant NRF-RSS2015-003, NRF-CRP24-2020-0002, and Hybrid Integrated Flexible Electronic System (HiFES) Prog. (hifes.nus.edu.sg), E6 Nanofab. Y-W.Z. acknowledges the support from Singapore A*STAR SERC CRF Award. Many thanks to A*STAR Computational Resource Centre and Singapore National Supercomputer Centre through the use of its high-performance computing facilities and Dr. Chai jianwei and Dr. Yu Bingxue for technical support.

## Author contributions

B.T., and A.V.-Y.T. conceived the project and wrote the paper. B.T., Y.L., J.F.L., and M.S. fabricated the RRAM devices. B.T. performed the MoS₂ exfoliation, UV-vis, Raman, PL, and electrical measurements. Z.G.Y. and Y.-W.Z. performed the DFT calculation. H.V. contributed to the CNN simulation and data analyses. E.Z. contributed to the AFM measurement and analysis. M.W. and J.W. performed STEM and data analysis. B.T., and A.V.-Y.T. provided input into the design of the experiments and the preparation of the manuscript.

## Competing interests

The authors declare no competing interests.
