## [Peer Review File · Nature Communications]

Title: Wafer-scale Solution-Processed 2D Material Analog Resistive Memory Array for Memory-Based ComputingREVIEWER COMMENTS

Reviewer #1 (Remarks to the Author):

Reviewer Letter to the Editor and Authors

Wafer-scale Solution-Processed 2D Material Analog Resistive Memory Array for Memory-Based Computing

Baoshan Tang¹, Yida Li¹, Hasita Veluri¹, Zhi Gen Yu³, Moaz Waqar², Jin Feng Leong¹, Maheswari Sivan¹, Evgeny Zamburg¹, Yong-Wei Zhang³, John Wang², Aaron V-Y Thean^{1*}

¹ Department of Electrical and Computer Engineering, National University of Singapore, Singapore 117576, Singapore,

² Department of Materials Science and Engineering, National University of Singapore, Singapore 117576, Singapore,

³ Institute of High Performance Computing, A*Star, Singapore 138632, Singapore.

The manuscript by Baoshan et al. presents a detailed investigation on a scalable process to produce analog resistive memory arrays. The authors describe a solution-based methodology to fabricate thin layers of MoS₂ with a good control on their size and thickness. The procedure adopted is certainly not new and already established in literature, however, the authors demonstrate to finely control and tune all the process parameter.

The MoS₂ thin films were characterized with appropriate spectroscopic techniques such as Raman, XPS, photoluminescence and Energy loss spectroscopy. Besides the material characterization, the authors conducted a deep and detailed investigation on the electrical characteristics for all the realized devices. Electrical performances were investigated as function of the temperature and flake lateral dimension, allowing the authors to develop a proper model for the inter-flake sulfur vacancies diffusion. Moreover, a detailed investigation on the dynamic response of the MoS₂ memristor was performed, enabling the training and learning of a MoS₂ memristor array, showing remarkable performance in terms of integration size, endurance and learning accuracy.

The recipe adopted for the MoS₂ memristor was then applied to realize a proof-of-concept 3-layer memory, demonstrating the possibility to use the adopted methodology to build monolithic 3D memory cubes, which is a promising strategy to co-integrate 2D materials in existing devices.

To conclude, I believe this manuscript is of high interest for the readership and should be accepted after minor revision once the few minor points listed below are addressed.

- Introduction pg. 2 raw 31: GPU as acronymous should be explicitly specified at least once

- Introduction pg. 2 raw 40: the authors should cite some recent references concerning the MoS₂ synthesis with scalable methodologies. Several methodologies from liquid solution and vacuum-based growth technologies were recently published. Just as example we list here few of them: Liu et al *Nanoscale Adv.*, (2021),3, 2117-2138; Timpel et al. *npj 2D Mater Appl* 5, 64 (2021); Nardi et al. *ACS Appl. Mater. Interfaces* (2018), 10, 40, 34392–34400, Lee et al. *ACS Appl. Nano Mater.* (2021), 4, 7, 6668–

6677.

- Introduction pg 3 raw 58: the authors should cite some references concerning the role of edge defects and their engineering.

- Introduction pg 3 raw 66: RS as acronym must be explicitly specified at least once

Along the entire manuscript: All the acronyms should be at least one specified for readers that are not expert in the field.

Reviewer #2 (Remarks to the Author):

This study reports that the solution processed MoS₂ can work as the memristive medium and demonstrates that a wafer-scale memristive device can be constructed. Although it is interesting to see wafer scale implementation memristive elements over solution processed MoS₂ thin film over wafer scale, the study needs a more critical analysis to demonstrate the merits of the current strategy.

1. For the memristor fabricated by the solution processed MoS₂, the authors claim that the vacancies at the edge of MoS₂ flakes dominate the conducting filament, and can be mediated through the intercalation processing. The operation voltage and I-V variation difference is not obvious for the memristors fabricated with MoS₂ flakes with different size. In this regard, it is essential to show more wafer statistics to demonstrate the observed difference is from the size effect other than any other factors.

2. The authors proposed an interesting but high speculative hypothesis to interpret the memristive effect. It is necessary to have more detailed structural characterizations, theory or exact interpretation of the size dependent behavior.

3. The single device performance, including the operation voltage, switching and retention characteristics there is no obvious advantage comparing with other works. The authors should carefully benchmark vs. the state of the art in the field, critically discussing the merits and limits of current design.

4. The authors present 10 memory state in Figure 4e. How reproducible in set and reset such memory states within a single device and across different devices. This is only meaningful if memory states can repeatedly and reproducibly set in a designable/programmable fashion. Based on the sharp I-V curve in Figure 3, it appears challenging practically set so many memory states?

5. About the CNN computing, there are many works implementing the image training based on a single device or based on integrated hardware structure. The authors need to more clearly discuss the potential benefit brought by the current implementation.

6. For the 3D integration strategy in this work, there is no any separation layers between different layer devices, also there is no obvious rectifying characteristics and on/off ratio is not high, how to avoid the crosstalk problems between different devices in different layers?

Reviewer #3 (Remarks to the Author):

The work presents a fabrication technique for memristor devices based on the spin coating of MoS₂ nanosheets on wafer scale. The devices show repeatable set/reset transitions in their I-V curves. The switching effect is attributed to the sulfur vacancy in the MoS₂ at the edges of the nanosheets. 3D

integration is shown by stacking multiple metal-MoS₂-metal layers.

Although interesting, the nanosheet approach cannot guarantee the necessary uniformity which is needed for scaling. The nanosheet size is typically in the micron/submicron range, therefore it might not be possible to scale the technology to the 10-20 nm range if switching takes place only at the nanosheet edge.

The application of these devices to neural networks is based on simulations with apparently optimistic assumptions about the precision of the device conductance, which is however not supported by data. The work seems therefore incomplete at least in terms of (i) demonstration of uniformity and scaling to the sub 100 nm scale, (ii) demonstration of the low variation to support the simulation results about the neural network.

See more comments in the following.

- The dependence of V_{set} and V_{reset} on the nanosheet size should be supported more rigorously. I suggest that at least 40-50 devices for each nanosheet size are measured, then the distribution of V_{set} and V_{reset} should be compared by clearly defining them (e.g., V_{reset} is the voltage corresponding to the maximum current under negative voltage). Finally, one should show V_{set} and V_{reset} as a function of the average nanosheet size, including error bars.
- Please specify the thickness of the spin-coated nanosheet layers obtained from various suspensions.
- Please explain the concept of temporal switching variations. The variation of switching characteristics from cycle to cycle are generally referred to as cycle-to-cycle variations.
- Please specify the electrode size in Fig. 3. Are the considerations about V_{set} and V_{reset} vs. nanosheet size still valid when the electrode size becomes comparable or smaller than the nanosheet? What is the scalability of the concept, given that the minimum average nanosheet density is 0.48 μm^2 ?
- There is a statement in the text: 'Unlike the stochastic formation of CF in amorphous oxide-based memristor, the solution processed MoS₂ memristor enables a more uniform RS characteristic due to the unique edge-confined VS conduction mechanism'. The edge-confined RS mechanism makes the present device less (instead of more) uniform than amorphous oxide-based memristors, where the structure is more uniform. In addition, memristors based on amorphous metal oxides are more scalable, since the device area is not limited by the nanosheet size.
- Please specify what is the nanosheet size of the devices in Fig. 4 used for CNN demonstration. What is the performance of the CNN for various nanosheet size?
- The linear and symmetric potentiation/depression in Fig. 4d would be useful for online training accelerators, rather than generic analog in-memory matrix multiplication. Anyway, the linearity and symmetry are required for a constant voltage and pulse width, contrary to the ramped voltage in the figure.
- The simulation results about the CNN accuracy in Fig. 4 might be misleading. The key parameter in controlling the performance is the precision of analogue conductance of the memristor device. However, it is not clear what precision was assumed and how many values of conductance were assumed in the device. The precision should be supported by experimental data about the cycle-to-cycle and device-to-device variation of conductance for a given program/erase algorithm. At first glance, given the large cycle-to-cycle variation of HRS and LRS in Fig. 3, it might be surprising that this device reaches a

precision close to floating point, which is presumably the precision of the GPU in Supplementary Table I.

- Please explain in Fig. 5 how were the devices in the first, second and third layer connected for the electrical measurements.

Point-by-Point Response to Reviewers

Reviewer #1: The manuscript by Baoshan et al. presents a detailed investigation on a scalable process to produce analog resistive memory arrays. The authors describe a solution-based methodology to fabricate thin layers of MoS₂ with a good control on their size and thickness. The procedure adopted is certainly not new and already established in literature, however, the authors demonstrate to finely control and tune all the process parameter.To conclude, I believe this manuscript is of high interest for the readership and should be accepted after minor revision once the few minor points listed below are addressed.

Response: We appreciate the reviewer's kind comments as well as useful suggestions. Please see below for our detailed responses.

Comment 1): Introduction pg. 2 raw 31: GPU as acronymous should be explicitly specified at least once.

Response: The full name of GPU-Graphics Processing Unit has been added in the Introduction Section (Page 2) in the manuscript.

Comment 2): Introduction pg. 2 raw 40: the authors should cite some recent references concerning the MoS₂ synthesis with scalable methodologies. Several methodologies from liquid solution and vacuum-based growth technologies were recently published. Just as example we list here few of them: Liu et al *Nanoscale Adv.*, (2021),3, 2117-2138; Timpel et al. *npj 2D Mater Appl* 5, 64 (2021); Nardi et al. *ACS Appl. Mater. Interfaces* (2018), 10, 40, 34392–34400, Lee et al. *ACS Appl. Nano Mater.* (2021), 4, 7, 6668–6677.

Response: We appreciate the suggestions. In response, we have included the following references in the Introduction Section in Page 2 with reference number 7-10 in the revised manuscript.

7 Liu, Y. & Gu, F. A wafer-scale synthesis of monolayer MoS₂ and their field-effect transistors toward practical applications. *Nanoscale Adv.* **3**, 2117-2138 (2021).

8 Timpel, M. et al. 2D-MoS₂ goes 3D: transferring optoelectronic properties of 2D MoS₂ to a large-area thin film. *npj 2D Mater. Appl.* **5**, 1-10 (2021).

9 Nardi, M. V. et al. Versatile and Scalable Strategy To Grow Sol–Gel Derived 2H-MoS₂ Thin Films with Superior Electronic Properties: A Memristive Case. *ACS Appl. Mater. Interfaces* **10**, 34392-34400 (2018).

10 Lee, C.-S. & Kim, T. H. Large-Scale Preparation of MoS₂/Graphene Composites for Electrochemical Detection of Morin. *ACS Appl. Nano Mater.* **4**, 7, 6668-6677 (2021).

Comment 3): Introduction pg 3 raw 58: the authors should cite some references concerning the role of edge defects and their engineering.

Response: Per your suggestion, we have cited the following references in Page 3 with reference number 23 to 27 in our revised manuscript. The following is a detailed description of the cited works.

Reference 23: V K. Sangwan, *et al.*, *Adv. Func. Mater.*, **2021**, 2107385

Sangwan *et al.* has reported the percolating networks of a diverse solution-processed 2D semiconductors for memristive switching. They explained that the thermally assisted electrical discharge that preferentially occur at the sharp edges of 2D nanosheets enables high switching ratios at low electrical fields.

Reference 24: C. Tan, *et al.*, *Chem. Soc. Rev.* 2015, 44, 2615

The authors have reviewed recent progress in the utilization of solution-processed 2D nanomaterials for non-volatile resistive memory devices. In this review, the authors have elaborated that the geometric dimensions of nanosheets and morphology of the percolating nanosheets network will affect the electrical switching behavior in solution-processed 2D materials based memristors.

Reference 25: S T. Han, *et al.*, *Adv. Sci.*, **2017**, 4(8): 1600435

The edge defects associated trap density in 2D black phosphorus quantum dots have been utilized to realize precise control over the SET voltages and conductance states of the RRAM.

Reference 26: S. Maheswari, *et al.*, *Nat. Commun.*, **2019**, 10(1): 1-12

In our previous publication in *Nature Communications*, we have reported low-voltage defects enabled RRAM by using aerosol jet printed WSe₂ nanosheets. In this work, we suggest that the key metrics in RRAM including endurance, switching speed and power can be potentially modulated through controlling the flake size together with defect density.

Reference 27: A. M. Abdelkader, *Nanoscale*, **2015**, 7, 6944–6956

The authors have elucidated the kinetics during the intercalation and exfoliation process of graphene, in which the edge sites and edge defect density can be affected by types of intercalants, solvents and values of applied voltage.

Comment 4) Introduction pg 3 raw 66: RS as acronym must be explicitly specified at least once.

Response: The acronym RS-resistive switching has already been specified at Page 3 Row 56.

Comment 5) Along the entire manuscript: All the acronyms should be at least one specified for readers that are not expert in the field.

Response: We have double checked that all the acronyms have been specified when they first show up in the manuscript and supplementary information.

Reviewer #2: This study reports that the solution processed MoS₂ can work as the memristive medium and demonstrates that a wafer-scale memristive device can be constructed. Although it is interesting to see wafer scale implementation memristive elements over solution processed MoS₂ thin film over wafer scale, the study needs a more critical analysis to demonstrate the merits of the current strategy.

Response: We thank the reviewer for the constructive suggestions. Please see below for the point-by-point response to your comments.

Comment 1): For the memristor fabricated by the solution processed MoS₂, the authors claim that the vacancies at the edge of MoS₂ flakes dominate the conducting filament, and can be mediated through the intercalation processing. The operation voltage and I-V variation difference is not obvious for the memristors fabricated with MoS₂ flakes with different size. In this regard, it is essential to show more wafer statistics to demonstrate the observed difference is from the size effect other than any other factors.

Response: Thanks for the helpful suggestion. As suggested, we have conducted the statistical analyses of the resistive switching characteristics over flake sizes. 50 devices from three batches of MoS₂ suspensions (suspension A, B and C) corresponding to different flake sizes are characterized. The results are shown in Supplementary Figure S12. As the flake size decreases from 2.40 μm to 0.48 μm , the average voltages needed for set and reset reduces from 1.52 V/-1.31V to 1.08 V/-1.05V. Moreover, there is a significant reduction in their standard derivation, indicating improved device-to-device uniformity. Our statistical analysis confirms the strong flake size dependent switching characteristics, which enables a novel material engineering parameter for the modulation of RRAM performance.

Supplementary Figure S12 has been added in the Supplementary Information.

The following description has been added in the Manuscript (Row 148, Page 8):

“The statistical analyses of 50 devices under different flake size further confirm the flake size dependent RS characteristics in MoS₂ memristor (Supplementary Figure S12).”

Supplementary Figure S12| Statistical analysis of the switching voltages as a function of MoS₂ nanosheets size. Histogram of the set voltage (V_{set}) and reset voltage (V_{reset}) for MoS₂ memristors made from suspension A (a,d), B (b,e) and C (c,f), respectively. To be noted, V_{reset} is defined as the voltage at the maximum current under negative bias. V_{set} is defined as the voltage where the current abruptly increases under positive bias. The statistical analysis of the standard deviation (σ) and average values of switching voltages of 50 devices in each batch reveal the improved switching uniformity and feasibility when shrinking the nanosheet size. Specifically, the RS happens at higher voltages of 1.52 V (set) and -1.31 V (reset) with a larger σ of 0.44 V and 0.26 V for MoS₂ suspension C. In contrast, a tighter distribution in V_{set} and V_{reset} together with smaller switching voltages has

been achieved when MoS₂ suspension A is used. This confirms the strong correlation between the flake geometry and the corresponding RS characteristics.

Comment 2): The authors proposed an interesting but high speculative hypothesis to interpret the memristive effect. It is necessary to have more detailed structural characterizations, theory or exact interpretation of the size dependent behavior.

Response: We appreciate the reviewer's constructive comments. The memristive effect that we proposed in this work has been well supported by detailed study that includes physical material study, device electrical characterizations and DFT calculations. In response to the reviewer's feedback, we recognize that these points may not have been articulated clearly in the paper. Therefore, we provide here an improved systematic presentation of the evidence to support our argument for a unique flake-edge defect percolation switching.

(1) First, we carefully characterized the defects responsible for the conductive path. Through XPS, the surface chemistry of the MoS₂ nanosheets indicates the sulfur deficiency (vacancy defects) that can contribute to the formation of conductive channels (Supplementary Figure S2). With KPFM and STEM, we characterized spatial distribution of sulfur vacancies (V_S) over the exfoliated MoS₂ flakes. We showed that high concentration of V_S are predominantly found at the flake edges (Supplementary Figure S6 and Figure 2 j, k). Furthermore, conductive AFM (C-AFM) conducted on a flake reveals that RS occur most readily near the MoS₂ flake edges as well. Thus, the C-AFM results provide direct supporting evidence that flake edges are more conductive due to elevated density of V_S, creating favorable regions for out-of-plane inter-flake RS conduction.

(2) Secondly, the observation of unique flake-size dependent switching characteristics of the RRAMs provides supporting evidence of edge-to-edge percolation conduction influenced by flake size modulation. We show that lower write voltages (to form the RS path between top and bottom RRAM electrodes) decreases as flake size diminishes, accompanied by tightening of the device-to-device switching voltages distribution. This is consistent with our DFT calculation, where smaller flakes associated with higher defect density cause the reduction in the energy barrier for V_S migration, thus enabling RS happens at low electrical field. The percolation path of V_S plays a key role in the

switching uniformity of MoS₂ memristor. For better clarity to the readers, a schematic diagram depicting the V_S percolation path under different flake size has been added as Supplementary Figure S11 (see below). The percolation paths (grey spheres) with larger flake size exhibit much higher randomness. In contrast, smaller MoS₂ flakes can provide a smoother and more uniform pathway for V_S percolation, offering well-controlled switching behavior in MoS₂ memristor.

(3) Thirdly, the fittings of *I-V* curves at HRS and LRS state to different transport model (Schottky, SCLC and Mott-VRH model) further shed light on the transport mechanism in our MoS₂ RRAM, demonstrating that the edge-to-edge percolation of V_S causes the forming and breaking of conductive filament and the dynamic modulation of Schottky barrier at MoS₂-contact interface.

Supplementary Figure S11| Sulfur vacancy percolation path at different nanosheet size. a. MoS₂ memristor with large nanosheet size. **b.** MoS₂ memristor with small nanosheet size. To be noted, the grey spheres represent V_S. The cycle-to-cycle variation depends on the uniformity of V_S percolation path in the our MoS₂ memristors. Large flake size has much higher randomness in the V_S percolation path, causing switching uniformity deterioration.

Comment 3): The single device performance, including the operation voltage, switching and retention characteristics there is no obvious advantage comparing with other works. The authors should carefully benchmark vs. the state of the art in the field, critically discussing the merits and limits of current design.

Response: We appreciate the reviewer for the constructive comment. As suggested, we have benchmarked with other 2D materials-based RRAM and also the state-of-the-art oxides-based RRAM in Supplementary Table II. As compared with other reported 2D counterparts, the merits of our MoS₂ RRAM include high endurance (10⁷ cycles), large number of memory states (50 states) and long retention (>10 years), all while with considerably low switching voltage (<1 V). Meanwhile, our work provides a practical way for the wafer-scale implementation of the 2D materials in a transfer-free manner, which shows great advantages over the chemical vapor deposition processing in terms of integration complexity. In addition, our MoS₂ RRAM offers the capability for RS modulation via flake size which is a process dimension that current oxides lack. All our processes are conducted at room-temperature, offering compelling compatibility with temperature-limited 3D monolithic integration as well as flexible electronics processing.

We agree that the RRAM performances like switching energy and device variations at the early stage of maturity, are not sufficiently good enough. But we believe that they can be further improved through materials, device and circuits co-optimization.

We have also updated the corresponding parts in the manuscript as below:

“Supplementary Table II benchmarks the overall performance metrics of different 2D materials-based memristors as well as the state-of-art oxide-based memristors. Our MoS₂ memristor shows the best performance in the integration size, endurance, learning accuracy and number of conductance states relative to other 2D materials. Especially, our proposed processes are transfer-free and room-temperature based, offering great compatibility with thermal budget limited 3D monolithic integration as well as flexible electronics. Moreover, our devices feature a new process dimension of performance modulation via flake size engineering, which is lacking in current oxide-based RRAMs. Despite limitations in high switching power and relatively large device variations of the current design, we believe there is still significant opportunity for improvement by materials, devices and circuits co-optimization in the future.” (Row 241, Page 12)

Supplementary Table II Benchmarking of our solution-processed MoS₂ RRAM devices with reported 2D materials-based RRAM and conventional oxides-based RRAM.

Active layer	V _{SET/RESET}	Forming voltage	Compliance current	Digital/ Analog	Conductance states	Endurance	Array size	Retention
Colloidal MoS ₂ ¹	5.5/-0.5V	Form-free	0.1 mA	digital	2	100	10×10	10 ⁴ s
Printed MoO _x -MoS ₂ ²	0.15/-0.1V	Form-free	0.5 mA	digital	2	50	single	8000 s
Aerojet printed MoS ₂ ³	0.18/-0.3 V	Form-free	1 mA	digital	6	200	single	10 ⁵ s
Graphene/MoS ₂ /Graphe ne ⁴	1.2/2.2 V	1.5V	0.7 mA	digital	2	10 ⁷	4×4	10 ⁵ s
Black phosphorus ⁵	3/-2.8 V	Form-free	1 mA	digital	4	-	single	10 ⁴ s
Graphene oxide ⁶	2.5/-2.5 V	Form-free	50 μA	analog	2	100	5×5	10 ⁵ s
CVD h-BN ⁷	2.65/-1.85 V	3~8.0V	1 μA	analog	26	8000	10×10	-
h-BN/Graphene/h-BN ⁸	4.0/3.8 V	5.0V	100 μA	digital	2	10 ⁶	12×12	10 ⁶ s

Pd/HfO _x /TiN ⁹	2.2/-2.2 V	5.25 V	100 μA	analog	2	>10 ⁸	single	10 ⁴ s
TaO _x /Ta ₂ O ₅ ¹⁰	-1.0/1.5 V	-	0.1 mA	analog	-	10 ⁷	single	10 ⁸ s
This work	0.65/-0.9V	Form-free	1 mA	analog	50	10⁷	10×10	10⁸ s

Comment 4): The authors present 10 memory state in Figure 4e. How reproducible in set and reset such memory states within a single device and across different devices. This is only meaningful if memory states can repeatedly and reproducibly set in a designable/programmable fashion. Based on the sharp I - V curve in Figure 3, it appears challenging practically set so many memory states?

Response: As shown in the I - V curves in Figure 3a-c. When applying negative bias on the MoS₂ RRAM, the device undergoes a gradual RESET process, implying multiple analog states. Therefore, we can reset the MoS₂ RRAM from the LRS states to varied intermediate HRS states. As shown in Figure 4d, the unchanged current measured during an elongated time of a single RRAM indicates the stable and non-volatile characteristics of those states.

In Supplementary Figure S19, we programmed a single MoS₂ RRAM to different analog states for 2 times, showing repeatable behavior.

Furthermore, we have demonstrated the multiple memory states in another four devices (Supplementary Figure S20 a-d). All of them can be programmed to multiple memory states, showing reproducible results.

The following Figures are added as Figure S19 and Figure S20 in the Supplementary Information.

The following description is added in Row 223, Page 11 in the Manuscript as below:

“The multiple memory states are programmable within single device, (Supplementary Fig. S19), and further demonstrated to be highly reproducible among different devices (Supplementary Fig. S20).”

Supplementary Figure S19| Repeatable programming of multiple memory states in MoS₂ memristor. *I-t* measured at 0.2 V for 10 memory states of (a) initially programmed and (b) reprogrammed MoS₂ memristor. “Fully set” refers to the memory state when device was fully set to LRS states by sweeping a positive bias to 2 V with compliance of 1 mA. After the MoS₂ RRAM was programmed from State 1 to State 10, it is fully set back to State 1 with 2V bias under 1 mA compliance (reprogram process). The value beside the curve refers to the reset voltage for each memory state. The repeatable memory states in the same device indicates those memory states are reproducible and programmable.

Supplementary Figure S20| Multiple memory states reproducible in different MoS_2 memristors. *I-t* measured at 0.2 V for 10 programmed memory states on four different MoS_2 RRAM devices.

Comment 5): About the CNN computing, there are many works implementing the image training based on a single device or based on integrated hardware structure. The authors need to more clearly discuss the potential benefit brought by the current implementation.

Response: In our presented implementation, the RRAM conductance curve is discretized into a finite number of states determined based on the observed device variability. The kernel elements are mapped onto these discrete states and stored within the RRAMs for processing. The unique aspects of our current implementation, which includes the conductance discretization based on variability and our floating-point mapping techniques, effectively combat device variability to reduce the system output error. We have previously implemented similar algorithm for oxide-based RRAM, ^{11,12} and in this work, we successfully demonstrate

system output error reduction with 2D materials in a memory array simulation. The system evaluation is based on variations captured through multi-devices characterizations (Supplementary Fig. S22), rather than single device data.

In addition to the above mentioned, for your convenience, more details of our system implementation can be found in the following.

We employ pulse width modulation to execute computations within the RRAMs. Owing to this, we reduce the power and area consumed by the periphery. We divide the RRAM conductance into 16 discrete states and use a 3-bit input feature matrix (IFM) resolution to execute the CNN within RRAMs. The high-accuracy computations that we perform here use low RRAM and IFM resolution to reduce system power/area compared to other works. A detailed description of the in-memory compute methodology used for the neural network simulations, along with its performance analysis, has been provided in our previous works.^{11,12}

Further, in our execution, we modeled the RRAM behavior based on the conductance curves of numerous devices over multiple cycles. We derived the variance and the mean of the measured data and used this to run our simulations. We modeled the variation at each discrete state as a gaussian function about the mean and derived the current at random from this Gaussian function in every iteration. Thus, the simulations performed in our work account for device irregularities such as limited conductance range and variability to provide an accurate estimate of the outputs. The conductance curves of 20 RRAMs used in this work, along with their discretization into 9 states, is provided in the figures below:

Supplementary Figure S22| Conductance modulation of MoS₂ memristors with device-to-device variability. In total, 20 memristors have been measured with 90 identical reset pulses. The conductance curves are divided into 9 states based on the demonstrated device-to-device variability.

For system analysis, a hysteron-based compact model, developed by Lehtonen et.al.¹³, has been calibrated to our MoS₂ RRAM as shown below:

Supplementary Figure S23| Calibration of the experimental data with Hysteron-based model. Comparison of developed spice model with experimental data shows good correlation.

The following text has been added to Note II of the Supplementary Information:

“The unique aspects of our CNN implementation, which includes conductance discretization based on variability and our floating-point mapping techniques, effectively combat device

variability to reduce the system output error. We have previously implemented similar algorithm for oxide-based RRAM,^{6,7} and in this work, we successfully demonstrate system output error reduction with 2D materials in a memory array simulation. The system evaluation is based on variations captured through multi-devices characterizations, rather than single device data (Supplementary Fig. S22).

We discretize the RRAM conductance curve into a finite number of states determined based on the observed device variability. The kernel elements are mapped onto these discrete states and stored within the RRAMs for processing. We show in our previous works that the conductance discretization based on variability and our mapping techniques effectively combat device variability to reduce the system output error. Also, we employ pulse width modulation to execute computations within the RRAMs to reduce the power and area consumed by the periphery. In our current implementation, we divide the RRAM conductance into 16 discrete states and use a 3-bit input image/4-bit kernel resolution to execute the CNN within RRAMs. The high-accuracy computations that we perform here using low resolution reduce system power/area compared to other works. A detailed description of our methodology, along with its performance analysis, has been provided in our previous works.^{6,7}

In our execution, we modeled the RRAM behavior based on the conductance curves of 20 devices over multiple cycles. We derived the variance/mean of the measured data and used this to run our simulations. We modeled the current variation at each discrete state as a gaussian function about the mean and derived the conductance at random from this function at every iteration. Thus, the simulations performed in our work account for device irregularities such as limited conductance range and variability to provide an accurate estimate of the output errors. The sample conductance curves of 20 RRAMs (cell size $5 \times 5 \mu\text{m}^2$) used in this work, along with their discretization into 9 states, is provided in the Figure S22. For system analysis, a hysteron-based compact model, developed by Lehtonen et.al.⁸, has been calibrated to our MoS₂ RRAM in Figure S23.”

Comment 6): For the 3D integration strategy in this work, there is no any separation layers between different layer devices, also there is no obvious rectifying characteristics and on/off ratio is not high, how to avoid the crosstalk problems between different devices in different layers?

Response: Thanks for the comments. We recognize that directly connecting the RRAM will lead to sneak current issue. This sneak current issue is common in 2D RRAM array. It requires careful control of the programing voltages. It also creates a sneak current problem in 3D RRAM stacks. One possible solution is to carefully choose the programing voltage schemes like one-half voltage bias scheme (The voltage is applied across the selected word and bit line, while all other bit and word lines are applied with a half voltage bias) or one-third voltage bias scheme. We understand this is not the optimal. The optimal solution is to integrate the selectors like transistors or diodes. As a proof of concept, we fabricated the 3-layer MoS₂ RRAM stacks in this work to demonstrate the feasibility of stacking these RRAM with little performance impact. A full view of the 3D stacked MoS₂ RRAM has been provided in Supplementary Information Figure S25. Each layer of the MoS₂ RRAM stacks can be accessible and programmable independently in which MoS₂ serves as the separation layers.

As for the successful implementation of the monolithic 3D memory arrays, it is necessary to have the inter-layer dielectric (ILD) layers between each layer of metal lines to isolate each layer devices and selectors to address the cross-talk issue. Although we didn't fabricate the ILD layer and selectors, our current method does not prevent us from integrating ILD and selectors with our MoS₂ RRAM in 3D monolithic arrays. In our previous work in *Nature Communication*,¹⁴ we have reported all WSe₂ 1T1R resistive RAM cell for monolithic 3D embedded memory integration, where the room-temperature processed WSe₂ transistor is utilized as selectors. In *2021 Symposia on VLSI technology and circuits*, we have demonstrated 3-D monolithically stack of 3-layer IGZO FETs, where the SiO₂ serves as the ILD layer.¹⁵ Notably, all those processes are room-temperature based and compatible with our MoS₂ memristors. Therefore, our proposed 3D integration strategy is achievable based on those well-developed techniques.

The following description has been added in Manuscript as below:

‘Based on our previous reports on M3D integration, interlayer dielectric between each layer of metal lines and selectors at each node of cross-point can be readily integrated in our monolithic 3D circuits to minimize the cross-talk issue and accurately program each memory cell.’ (Row 281, Page 14)

Supplementary Figure S25| 3D stacked MoS₂ memristors. **a.** The schematic diagram and corresponding SEM images (**b-c**) of the monolithic 3D MoS₂ RRAM stacks. **d-f.** Equivalent circuits representing the voltage scheme used for addressing selected memristor in the bottom, middle and top layer, respectively. C1, C2, C3 and C4 denote the metal contacts for each layer. The 3D MoS₂ stacks consist of C1/MoS₂/C2 (bottom layer), C2/MoS₂/C3 (middle layer) and C3/MoS₂/C4 (top layer), respectively. Each layer of the MoS₂ RRAM can be accessible and programmable independently in which MoS₂ serves as the separation layer.

Reviewer #3: The work presents a fabrication technique for memristor devices based on the spin coating of MoS₂ nanosheets on wafer scale. The devices show repeatable set/reset transitions in their I-V curves. The switching effect is attributed to the sulfur vacancy in the MoS₂ at the edges of the nanosheets. 3D integration is shown by stacking multiple metal-MoS₂-metal layers. Although interesting, the nanosheet approach cannot guarantee the necessary uniformity which is needed for scaling. The nanosheet size is typically in the micron/submicron range, therefore it might not be possible to scale the technology to the 10-20 nm range if switching takes place only at the nanosheet edge. The application of these devices to neural networks is based on simulations with apparently optimistic assumptions about the precision of the device conductance, which is however not supported by data. The work seems therefore incomplete at least in terms of (i) demonstration of uniformity and scaling to the sub 100 nm scale, (ii) demonstration of the low variation to support the simulation results about the neural network.

Response: We thank the reviewer for the important comments. The size of the solution processed MoS₂ nanosheets can be tuned over a large range from several microns to tens of nanometers. Therefore, the uniform switching can be guaranteed when the device is scaled down to tens of nanometers. We have shown a 100 nm×100 nm RRAM with fully functional resistive switching when the nanosheet size is also scaled down to comparable range (please see Response to Comment 4). As detailed in our response to Reviewer #2, we conducted our simulations on the measured RRAM conductance curves. We modeled the RRAM behavior based on the conductance curves of 20 devices over multiple pulses. Instead of using the continuous conductance curve for CNN execution, we discretized the finite curve into limited states based on observed variability. We derived the variance and the mean of the measured data and then used this to run our simulations. We modeled the variation at each discrete state as a Gaussian function about the mean and derived the current at random from this Gaussian function in every iteration. Owing to the conductance discretization and incorporation of observed device-to-device variability into our simulations, our work account for these devices irregularities to provide an accurate estimate of the outputs. A detailed description of the in-memory computing methodology used for the neural network simulations, along with its

performance analysis, has been provided in our previous work.^{11,12} The corresponding information has been added to Note II in the Supplementary Information. The comments about the uniformity, scaling and CNN simulation have been further addressed in the following response.

Comment 1) The dependence of V_{set} and V_{reset} on the nanosheet size should be supported more rigorously. I suggest that at least 40-50 devices for each nanosheet size are measured, then the distribution of V_{set} and V_{reset} should be compared by clearly defining them (e.g., V_{reset} is the voltage corresponding to the maximum current under negative voltage). Finally, one should show V_{set} and V_{reset} as a function of the average nanosheet size, including error bars.

Response: We appreciate the reviewer for this constructive comment. Following the reviewer's suggestion, 50 devices for each nanosheet size have been measured and their distribution of V_{set} and V_{reset} are plotted in Supplementary Information Figure S12. The definition of V_{set} and V_{reset} is added in the caption of Figure S12.

Supplementary Figure S12| Statistical analysis of the switching voltages as a function of MoS₂ nanosheets size. Histogram of the set voltage (V_{set}) and reset voltages (V_{reset}) from MoS₂ suspension A (**a, d**), B (**b, e**) and C (**c, f**), respectively. To be noted, V_{reset} is defined as the voltage at the maximum current under negative bias. V_{set} is defined as the voltage where the current abruptly increases under positive bias.

Comment 2) Please specify the thickness of the spin-coated nanosheet layers obtained from various suspensions.

Response: We have added the AFM step profile of the MoS₂ thin films from various suspensions in Supplementary Information Figure S4.

Supplementary Figure S4 | Optical image and corresponding AFM images of the spin coated MoS₂ film after patterning and etching. **a.** Optical images; **b-d.** AFM images of MoS₂ films from Suspension C, B and A, respectively. The inset is the step profile measured at the edge of the MoS₂ film. The square-shaped MoS₂ thin film was patterned and etched to measure the film thickness. Three types of MoS₂ suspensions result in similar thickness with excellent coverage.

The corresponding text has also been updated in the Manuscript as below:

“... achieving a smooth and thickness ranging from 10.5 to 11.4 nm (Fig. 2a, Supplementary Fig. S3, 4).” (Row 93, Page 5)

Comment 3) Please explain the concept of temporal switching variations. The variation of switching characteristics from cycle to cycle are generally referred to as cycle-to-cycle variations.

Response: The temporal switching variations in this work refers to “cycle-to-cycle variations”. We have replaced the term “temporal switching variations” with “cycle-to-cycle” variations to avoid confuse.

Comment 4) Please specify the electrode size in Fig. 3. Are the considerations about V_{set} and V_{reset} vs. nanosheet size still valid when the electrode size becomes comparable or smaller than the nanosheet? What is the scalability of the concept, given that the minimum average nanosheet density is 0.48 μm^2 ?

Response: The electrode size of 5 $\mu\text{m} \times 5 \mu\text{m}$ has been added in the caption of Figure 3. To further elaborate the switching mechanism, we have fabricated scaled devices (100 nm \times 100 nm)

with average nanosheet size of 0.48 μm . Owing to the edge switching effect, the scaled devices shows negligible resistive switching effect with very small memory window (please see below Figure S16 b). Therefore, when the electrode size becomes much smaller than nanosheets, there are not enough edge sites or defects within the electrode region to facilitate the resistive switching. This verifies the edge-dominant switching mechanism in our MoS₂ RRAM. However, we do not see any roadblocks that could prevent the scaling of our RRAM devices since the size of MoS₂ flakes are also tunable within a wide range. To prove this, we have prepared another MoS₂ suspension with average flake size of 0.12 μm (Supplementary Figure S16a). The uniform bipolar resistive switching behavior are preserved in the scaled devices (Supplementary Figure S16c). The concept is therefore, still valid given that the flake size is scalable.

Supplementary Figure S16| Scaled MoS₂ memristor. **a.** The lateral flake size distribution of MoS₂ suspension D. Scaled MoS₂ memristor (100 nm×100 nm) made from **(b)** MoS₂ suspension A ($\lambda = 0.48 \mu\text{m}$) and **(c)** suspension D ($\lambda = 0.12 \mu\text{m}$). λ is the average nanosheet size. Owing to the edge switching effect, the scaled device shows negligible RS effect when large nanosheet size was used as shown in Figure S16b. There are not enough edge sites or defects within the electrode region to facilitate the resistive switching. However, when MoS₂ nanosheet size reduces to around 0.12 μm , uniform bipolar resistive switching behavior are observed in the scaled devices (Figure S16c). It verifies the scalability of the proposed concept since the MoS₂ nanosheet size is also tunable within a wide range.

The following description is added in the Row 202, Page 10, Manuscript as below:

‘The uniform bipolar RS behavior can also be preserved when the device size is reduced to 100 nm×100 nm (Supplementary Fig. S16), demonstrating good scalability of the proposed process.’

The method for the preparation of suspension D has been added in Methods Section in the Manuscript as below:

“To prepare suspension D, suspension A was further sonicated, followed by centrifuge at 5000 r.p.m for 3 min. The supernatant was collected and named suspension D.”

Comment 5) There is a statement in the text: ‘Unlike the stochastic formation of CF in amorphous oxide-based memristor, the solution processed MoS₂ memristor enables a more uniform RS characteristic due to the unique edge-confined V_S conduction mechanism’. The edge-confined RS mechanism makes the present device less (instead of more) uniform than amorphous oxide-based memristors, where the structure is more uniform. In addition, memristors based on amorphous metal oxides are more scalable, since the device area is not limited by the nanosheet size.

Response: Thanks for the comment. Due to uncontrollable ion transport dynamics, the amorphous oxide-based memristors often suffer from stochastic filament formation through defects, resulting in non-uniform and inconsistent resistive switching.^{16,17} Therefore, several strategies have been reported to confine the filament to reduce the device variations. Choi *et al* have demonstrated enhanced switching uniformity through confinement of the conductive filament into one-dimensional threading dislocations in single-crystalline SiGe epitaxial layer.¹⁶ Very recently, Li *et al* has reported that the defective hetero-phase grain boundaries in 2D PdSe₂ memristors can guide the formation of conductive filaments, resulting in a six-fold improvement in switching variations.¹⁷ Those two works provide important ways to address the stochastic filament formation faced with amorphous oxide-based memristors. In our work, the vacancies percolation path may play a special role to modulate the growth of the filament, thus providing us additional degree to optimize the properties of RRAM by tuning the defective MoS₂ flake size. The switching uniformity in our MoS₂ memristors depends on two factors, namely the

flake size and the alignment of flakes. Although the flake size can be tuned via engineering approach during materials preparation, there is no efficient way for perfect alignment of flakes. Therefore, we agree that the vacancies percolation paths or filaments are still stochastic in MoS₂ memristor. But the unique V_S conduction mechanism and the ability to control the size distribution and edge defects density of the MoS₂ flakes give us an efficient pathway for RS modulation, which is not achievable in oxide-based RRAM.

To avoid confusion about this sentence, and have a better understanding on the advantage of our work, we would like to change the sentence to ‘Unlike the stochastic formation of CF in amorphous oxide-based memristor, the solution processed MoS₂ memristor enables **a better control over** RS characteristic due to the unique edge-confined V_S conduction mechanism’.

As response to Comment 4, the scalability of the proposed work wouldn’t be an issue because the MoS₂ flake size is tunable over a large range from a few microns to tens of nanometers.

Comment 6) Please specify what is the nanosheet size of the devices in Fig. 4 used for CNN demonstration. What is the performance of the CNN for various nanosheet size?

Response: Thanks for the comment. The nanosheet size of the devices in Fig. 4 is 0.48 μm. The size of the nanosheets has been specified in the caption of Fig. 4. CNN performance mainly depends on the non-linearity of devices, conductance range and the variability exhibited by RRAM devices. We determined the DC conductance range and the confidence intervals for the conductance spread using the bootstrap method in **R**. We have documented the results in Supplementary Figure S24 below. The figure delineates that the memory window increases progressively from suspension A to C. While the conductance spread does not follow a pattern, it is <10% of the mean for all three suspension types considered. Thus, our CNN simulations, which we performed with a σ/μ of 0.35 on devices fabricated using suspension A (5×5 μm² device size), account for the worst-case scenario. Furthermore, as explained previously, our in-memory compute methodology accounts for device variability issues, thereby preventing accuracy degradation. Hence, the CNN classification accuracy would remain unaltered for different nanosheet sizes.

Supplementary Figure S24| HRS and LRS current distribution of MoS₂ RRAM as a function of nanosheet size. The current of HRS and LRS current was read at 0.6 V.

Figure S24 and the following text have been added to Supplementary Note II:

“CNN performance depends on the non-linearity, conductance range, and variability exhibited by the RRAM devices. Figure S22 illustrates that our RRAM devices demonstrate a $2\times$ linear conductance change over 90 reset pulses. Hence, we determined the memory window and the confidence intervals for the conductance spread of devices fabricated with different suspensions using the bootstrap method in **R**. We documented the results in Figure S24. The figure delineates that the memory window increases progressively with nanosheet size. While the conductance spread does not follow a pattern, it is $<10\%$ of the mean for all nanosheet sizes considered. Thus, our CNN simulations, which we performed with a σ/μ of 0.35 on devices fabricated with nanosheet size of 0.48 μm , account for the worst-case scenario. Furthermore, as explained previously, our in-memory compute methodology accounts for device variability issues, thereby preventing accuracy degradation. Hence, the CNN classification accuracy would remain unaltered for different nanosheet sizes.”

Comment 7) The linear and symmetric potentiation/depression in Fig. 4d would be useful for online training accelerators, rather than generic analog in-memory matrix multiplication.

Anyway, the linearity and symmetry are required for a constant voltage and pulse with, contrary to the ramped voltage in the figure.

Response: The symmetric potentiation/depression with increased voltage pulses in Fig. 4e might be a bit misleading. Actually, constant pulse scheme (-1.7V/100ns) was used in the depression for CNN simulation (See Figure S22, Supplementary Information). We totally agree that the linear and symmetry potentiation/depression of conductance states are useful for online training accelerators. We will work on leveraging on this advantage and develop efficient training accelerators in the future.

Supplementary Figure S22| Conductance modulation of MoS₂ memristors with device-to-device variability. In total, 20 memristors have been measured with 90 identical reset pulses. The conductance curves are divided into 9 states based on the demonstrated device-to-device variability.

Comment 8) The simulation results about the CNN accuracy in Fig. 4 might be misleading. The key parameter in controlling the performance is the precision of analogue conductance of the memristor device. However, it is not clear what precision was assumed and how many values of conductance were assumed in the device. The precision should be supported by experimental data about the cycle-to-cycle and device-to-device variation of conductance for a given program/erase algorithm. At first glance, given the large cycle-to-cycle variation of HRS and

LRS in Fig. 3, it might be surprising that this device reaches a precision close to floating point, which is presumably the precision of the GPU in Supplementary Table I.

Response: We thank the reviewer for this comment. For the CNN demonstration, we divided the RRAM conductance curve into 16 states and assumed a 3-bit input pulse resolution. As explained in our response to Comment 5 raised by reviewers #2, we modeled the RRAM behavior based on the conductance curves of 20 devices over multiple cycles (Supplementary Figure S22). We derived the variance and the mean of the measured data and used this to run our simulations. The maximum derived variance (σ/μ) at any state is 0.35. We modeled the variation at each discrete state as a gaussian function about the mean and derived the current at random from this Gaussian function in every iteration.

Our in-memory compute methodology discretizes conductance and incorporates the observed device-to-device variability into our simulations. We discretize the conductance curve based on the observed current range and the conductance variability that the devices demonstrate (Supplementary Fig. S22). Owing to this, our method accounts for these device irregularities to improve system accuracy. A detailed description of the in-memory compute methodology used for the neural network simulations, along with its error and performance analysis, has been provided in our previous works.^{11,12} Our variation-tolerant in-memory algorithm, in conjunction with the high-yield device characteristics, resulting in the reported accuracy. Information regarding the same has been added to Note II of the Supplementary Information.

Comment 9) Please explain in Fig. 5 how were the devices in the first, second and third layer connected for the electrical measurements.

Response: A full view of the 3D stacked RRAM is shown in Supplementary Figure S25 as below. In our design, the bottom layer of MoS₂ RRAM is made of metal contact 1 (C1)/MoS₂/metal contact 2 (C2). The middle layer is made of C2/MoS₂/C3. The top layer is made of C3/MoS₂/C4. Therefore, each layer of the MoS₂ RRAM stacks can be accessible and programmable independently in which MoS₂ serves as the separation layers. For example, the bottom RRAM is selected with applying a bias of V on electrode C2 while keeping C1 grounded. C3, and C4 will share the same voltage as C2, thus no current flow through C3 and C4.

The following description has been added in Manuscript as below:

‘The overview of 3D stacked MoS₂ memristor and the corresponding electrical measurements can be found in Supplementary Fig. S25.’ (Row 264, Page 13)

Supplementary Figure S25| a. The schematic diagram and corresponding SEM images (**b-c**) of the monolithic 3D MoS₂ RRAM stacks. **d-f.** Equivalent circuits representing the voltage scheme used for addressing selected memristor in the bottom, middle and top layer, respectively. C1, C2, C3 and C4 denote the metal contacts for each layer. The 3D MoS₂ stacks consist of C1/MoS₂/C2 (bottom layer), C2/MoS₂/C3 (middle layer) and C3/MoS₂/C4 (top layer), respectively. Each layer of the MoS₂ RRAM can be accessible and programmable independently in which MoS₂ serves as the separation layer. For example, the bottom RRAM is selected when a voltage bias of V is applied on electrode C2 with C1 grounded. C3, and C4 are given the same voltages as C2, thus no current flow through C3 and C4 due to zero voltage potential drops.

Reference

- 1 Son, D. *et al.* Colloidal synthesis of uniform-sized molybdenum disulfide nanosheets for wafer-scale flexible nonvolatile memory. *Adv. Mater.* **28**, 9326-9332 (2016).
- 2 Bessonov, A. A. *et al.* Layered memristive and memcapacitive switches for printable electronics. *Nat. Mater.* **14**, 199-204 (2015).
- 3 Feng, X. *et al.* A fully printed flexible MoS₂ memristive artificial synapse with femtojoule switching energy. *Adv. Electron. Mater.* **5**, 1900740 (2019).
- 4 Wang, M. *et al.* Robust memristors based on layered two-dimensional materials. *Nat. Electron.* **1**, 130-136 (2018).
- 5 Han, S. T. *et al.* Black phosphorus quantum dots with tunable memory properties and multilevel resistive switching characteristics. *Adv. Sci.* **4**, 1600435 (2017).
- 6 Jeong, H. Y. *et al.* Graphene oxide thin films for flexible nonvolatile memory applications. *Nano Lett.* **10**, 4381-4386 (2010).
- 7 Chen, S. *et al.* Wafer-scale integration of two-dimensional materials in high-density memristive crossbar arrays for artificial neural networks. *Nat. Electron.* **3**, 638-645 (2020).
- 8 Sun, L. *et al.* Self-selective van der Waals heterostructures for large scale memory array. *Nat. Commun.* **10**, 1-7 (2019).
- 9 Wu, Q. *et al.* Improvement of durability and switching speed by incorporating nanocrystals in the HfO_x based resistive random access memory devices. *Appl. Phys. Lett.* **113**, 023105 (2018).
- 10 Lee, S. R. *et al.* Multi-level switching of triple-layered TaO_x RRAM with excellent reliability for storage class memory. 2012 Symposium on VLSI Technology (VLSIT). 71-72 (IEEE).
- 11 Veluri, H., Li, Y., Niu, J. X., Zamburg, E. & Thean, A. V.-Y. High-throughput, area-efficient, and variation-tolerant 3-D in-memory compute system for deep convolutional neural networks. *IEEE Internet Things J.* **8**, 9219-9232 (2021).
- 12 Veluri, H., Chand, U., Li, Y., Tang, B. & Thean, A. V.-Y. A low-power DNN accelerator enabled by a novel staircase RRAM array. *IEEE Trans. Neural Netw. Learn. Syst.* (2021).

- 13 Lehtonen, E. & Laiho, M. in *2010 12th International Workshop on Cellular Nanoscale Networks and their Applications (CNNA 2010)*. 1-4 (IEEE).
- 14 Sivan, M. *et al.* All WSe₂ 1T1R resistive RAM cell for future monolithic 3D embedded memory integration. *Nat. Commun.* **10**, 1-12 (2019).
- 15 Chand, U. *et al.* 2-kbit array of 3-D monolithically-stacked IGZO FETs with low SS-64mV/dec, ultra-low-leakage, competitive μ -57 cm²/Vs performance and novel nMOS-only circuit demonstration. *2021 Symposium on VLSI Technology*. 1-2 (IEEE, 2021).
- 16 Choi, S. *et al.* SiGe epitaxial memory for neuromorphic computing with reproducible high performance based on engineered dislocations. *Nat. Mater.* **17**, 335-340 (2018).
- 17 Li, Y. *et al.* Anomalous resistive switching in memristors based on two-dimensional palladium diselenide using heterophase grain boundaries. *Nat. Electron.* **4**, 348-356 (2021).

REVIEWERS' COMMENTS

Reviewer #1 (Remarks to the Author):

Reviewer Letter to the Editor and Authors

Wafer-scale Solution-Processed 2D Material Analog Resistive Memory Array for Memory-Based Computing

Baoshan Tang¹, Yida Li¹, Hasita Veluri¹, Zhi Gen Yu³, Moaz Waqar², Jin Feng Leong¹, Maheswari Sivan¹, Evgeny Zamburg¹, Yong-Wei Zhang³, John Wang², Aaron V-Y Thean^{1*}

The authors addressed all the minor comments highlighted in the first revision, after revision I can conclude that this manuscript is of high interest for the readership and should be accepted in the present status for publication in Nat. Comm.

Reviewer #3 (Remarks to the Author):

The work has been significantly revised and improved. However, there are still a couple of open issues that need to be further improved.

- Supplementary Figure S24, please clarify the unit of the nanosheet size.
- Supplementary Figure S25 is misleading since it shows a stack of three devices only, instead of a device array that is needed to compute the vector summation and multiplication. Please provide a scheme for selecting/unselecting devices in a crosspoint array, instead of an individual device. Also note that the applied voltage should be different for every device, which is needed for the multiplication of a voltage vector times a conductance vector.

Point-by-Point Response to Reviewers

Reviewer #3: The work has been significantly revised and improved. However, there are still a couple of open issues that need to be further improved.

Response: We appreciate the reviewer's kind comments as well as useful suggestions. Please see below for our detailed responses.

Comment 1) Supplementary Figure S24, please clarify the unit of the nanosheet size.

Response: Thanks. We have clarified the unit of the nanosheet size in Supplementary Figure S24.

Comment 2) Supplementary Figure S25 is misleading since it shows a stack of three devices only, instead of a device array that is needed to compute the vector summation and multiplication. Please provide a scheme for selecting/unselecting devices in a crosspoint array, instead of an individual device. Also note that the applied voltage should be different for every device, which is needed for the multiplication of a voltage vector times a conductance vector.

Response: Per your suggestion, we have added a new figure below (see Supplementary Figure S26) to illustrate the 3D memory in a crossbar array. The detailed voltage schemes used for selecting/unselecting the memory cell and performing in-memory matrix multiplication are described in the Figure S26 b and c, respectively. The following sentence is added in the Manuscript.

"As an example, the M3D integrated 1T1R arrays with corresponding circuit diagram are illustrated in Supplementary Fig. S26." (Line 286, Page 14)

Supplementary Figure S26 | Illustration of monolithic 3D integration of one transistor one

RRAM (1T1R) array. a. The schematic diagram of a three-layered 1T1R M3D arrays where

MoS₂ memristor is in series with a selection transistor to eliminate the sneak path current. The

inset showing the detailed structure of the memory cell. SL, BL and WL represent source line,

bit line and world line. The gates of transistor and top electrodes of RRAM are connected by the

WL and BL, respectively. The source of the transistor is connected to the SL. **b.** Equivalent

circuits showing the voltage scheme used for addressing selected RRAM. WL serves as the

selecting line. Signal inputs are applied to the BL, while signal outputs are collected at SL. **c.**

Equivalent circuits showing the voltage scheme used for performing in-memory matrix

multiplication within the RRAM arrays.

To select one cell, different WL voltages are used for SET and RESET (Fig. S26b). For example, for SET, a small voltage is applied on WL to turn on the selection transistor, while BL voltage is applied to set the state. For RESET, a large voltage is applied on WL to turn on the selection transistor while compensating the voltage drop on the RRAM cell. Meanwhile, SL voltage is applied to reverse the current to reset the RS state in a typical bipolar RRAM. For the implementation of in-memory matrix multiplication, kernel elements are stored within the RRAMs as conductance levels. Information is stored in the time period of the voltage pulses applied at BL. Since input pulses are applied to all the RRAMs in the array simultaneously, V_{DD} is applied at the WL of all transistors. Signal outputs are collected at SL (Fig. S26c).